# Integrated Variational Fourier Features for Fast Spatial Modelling with Gaussian Processes

**Talay M Cheema**  *tmc49@cam.ac.uk*
*Department of Engineering*
*University of Cambridge*

**Carl Edward Rasmussen**
*Department of Engineering*
*University of Cambridge*

Reviewed on OpenReview: *https://openreview.net/forum?id=PtBzWCaCYB*

## Abstract

Sparse variational approximations are popular methods for scaling up inference and learning in Gaussian processes to larger datasets. For $N$ training points, exact inference has $O(N^3)$ cost; with $M \ll N$ features, state of the art sparse variational methods have $O(NM^2)$ cost. Recently, methods have been proposed using more sophisticated features; these promise $O(M^3)$ cost, with good performance in low dimensional tasks such as spatial modelling, but they only work with a very limited class of kernels, excluding some of the most commonly used. In this work, we propose integrated Fourier features, which extends these performance benefits to a very broad class of stationary covariance functions. We motivate the method and choice of parameters from a convergence analysis and empirical exploration, and show practical speedup in synthetic and real world spatial regression tasks.

## 1 Introduction

Gaussian processes (GPs) are probabilistic models for functions widely used in machine learning applications where predictive uncertainties are important – for example, in active learning, Bayesian optimisation, or for risk-aware forecasts (Rasmussen & Williams, 2006; Hennig et al., 2022; Garnett, 2023). The hyperparameters of these models are often learnt by maximising the marginal likelihood, so it is important that this quantity can be evaluated fairly cheaply, especially to facilitate comparison of multiple models, or multiple random restarts for robustness. Yet, for $N$ datapoints, the time cost is $O(N^3)$, associated with calculating the precision matrix of the data and its determinant. This is prohibitively large for many datasets of practical interest. A particularly important class of problems is spatial modelling, where often Gaussian process regression is applied in low (2-4) dimensions to very large datasets, and ideally the choice of prior process, encapsulated by the prior covariance function, is guided by domain-specific knowledge.

One popular approach for improving scalability is to use a sparse variational approximation, wherein $M < N$ inducing features are used as a compact representation, and a lower bound of the log marginal likelihood is maximised. In the conjugate setting, where the measurement model is affine with additive, white, and Gaussian noise, the variationally optimal distribution of the inducing features is available in closed form (SGPR; Titsias, 2009). This reduces the size of the precision matrix from $N \times N$ to $M \times M$, but in practice multiplications by the cross covariance matrix between the data and the features dominates the computation cost at $O(NM^2)$.

SGPR with inducing points – where the features are point evaluations of the latent function – can be interpreted as replacing data measured with independent and identical noise with a reduced pseudo-dataset measured at different locations and with correlated and variable noise. This pseudo-dataset is selected to minimise distortion in the posterior distribution over functions (or, equivalently, minimum discrepancy

between the lower bound and the log marginal likelihood). This is particularly advantageous in the common case that the data is oversampled, where it is possible to set $M \ll N$ with asymptotically vanishing distortion (Burt et al., 2019; 2020a). Inducing points are the state of the art solution, but the scaling with $N$ is problematic. One popular way to avoid this is to use batches of data (SVGP; Hensman et al., 2015). But this necessitates the use of stochastic, typically first order, optimisers; in the conjugate setting, this leads to iterative learning of the variational distribution which is otherwise available in closed form.

Ideally, we would like to find an approximate inference method which avoids the $O(N)$ scaling for *any* dataset and *any* prior by careful design of the inducing features. But this is more generality than we can reasonably expect, and methods are generally restricted in the prior covariance functions they support, which in turn restricts the freedom of modellers. Existing work on zonal kernels on spherical domains and tensor products of Matérn kernels on rectangular subsets of $\mathbb{R}^D$ give a recipe for taking the $O(N)$ part of the computation outside of the optimisation loop for low dimensional datasets (Dutordoir et al., 2020; Hensman et al., 2017; Cunningham et al., 2023).

In this work, we propose integrated variational Fourier features (IFF), which provide the same computational benefits, but for a much broader class of sufficiently regular stationary kernels on $\mathbb{R}^D$.[1] We achieve this by allowing for modest numerical approximations in the evaluation of the learning objective and posterior predictives, rather than searching for mathematically exact methods. Yet, in contrast to those previous approaches, we also provide convergence guarantees. This both provides reassurance that the number of features required scales well with the size of the training data, and shows that the numerical approximations do not significantly compromise performance.

In Section 2 we review variational GP regression in the conjugate setting, and we review related work in Section 3. In Section 4 we present our IFF method, and the complexity analysis; the main convergence results and guidance for tunable parameter selection follows in Section 4.1. Finally in Section 5 we evaluate our method experimentally, showing significant speedup relative to SGPR in low dimensions, and competitive performance compared to other fast methods, with broader applicability. A summary of our contributions is as follows.

- We present a new set of variational features for Gaussian process regression, whose $O(M^2)$ memory cost and $O(M^3)$ per optimisation step computational cost greatly increases scalability for low dimensional problems compared to standard approaches – demonstrated on large scale regression datasets – and can be applied using a broad class of stationary covariance functions on $\mathbb{R}^D$.

- We provide converge results demonstrating the number of features required for an arbitrarily good approximation to the log marginal likelihood grows sublinearly for a broad class of covariance functions.

- We provide reasonable default choices of parameters in our algorithm, including the number of inducing features $M$, based on an empirical study and motivated by our theoretical results.

## 2 Background

In the conjugate setting, the probabilistic model for Gaussian process regression is

$$f \sim \mathcal{GP}(0, k) \quad y_n = f(x_n) + \rho_n \quad \rho_n \sim \mathcal{N}(0, \sigma^2) \tag{1}$$

for $n \in \{1 : N\}$, with each $x_n \in \mathbb{R}^D$ and $\rho_n, y_n \in \mathbb{R}$, with the covariance function or kernel $k : \mathbb{R}^D \times \mathbb{R}^D \to \mathbb{R}$ symmetric and positive definite. Let $\mathfrak{f} = [..., f(x_n), ...]^\top$, and $K_{ab}$ be the covariance matrix of finite dimensional random variables $a, b$. For example, $K_{\mathfrak{ff}}$ is the $N \times N$ matrix with $[K_{\mathfrak{ff}}]_{nn'} = k(x_n, x_{n'})$. The posterior predictive at some collection of inputs $x_*$ and marginal likelihood are as follows, where $x = x_{1:N}, y = y_{1:N}$, and $K_{*\mathfrak{f}}$ is defined analagously to $K_{\mathfrak{ff}}$. Then, the posterior predictive distribution and

---

[1]We assume the kernel's spectral density has bounded second derivative.

marginal likelihood are as follows (Rasmussen & Williams, 2006, Chapter 2).

$$p(f(x_*)|x, y) = \mathcal{N}(f(x_*)|K_{*\mathfrak{f}}(K_{\mathfrak{f}\mathfrak{f}} + \sigma^2 I)^{-1} y, K_{**} - K_{*\mathfrak{f}}(K_{\mathfrak{f}\mathfrak{f}} + \sigma^2 I)^{-1} K_{\mathfrak{f}*}) \tag{2}$$

$$p(y|x) = \mathcal{N}(y|0, K_{\mathfrak{f}\mathfrak{f}} + \sigma^2 I) \quad \mathcal{L} = \log p(y|x) \tag{3}$$

We optimise the latter with respect to the covariance function's parameters. The data precision matrix $A = (K_{\mathfrak{f}\mathfrak{f}} + \sigma^2 I)^{-1}$, which depends on the value of the hyperparameters, dominates the cost, as for each evaluation of the log marginal likelihood $\mathcal{L}$, we need to compute its log determinant and the quadratic form $y^\top A y$, both of which incur $O(N^3)$ computational cost in general. Note that the posterior predictive is the prior process conditioned on $y = \mathfrak{f} + \rho$.

For the variational approximation, we construct an approximate posterior[2] $q(f) = \int p(f|u)q(u)du \approx p(f|y)$ where $u = u_{1:M}$ is a collection of inducing features with prior distribution $p(u)$. That is, we condition on $u$ instead of $y$ and average over an optimised distribution on $u$. The classic choice is inducing points, where $u_m = f(z_m)$ for some $z_m \in \mathbb{R}^D$. We maximise a lower bound on the log marginal likelihood ($D_{KL}$ is the KL divergence).

$$\mathcal{F} = \int q(f) \log \frac{p(y, f, u)}{q(f)} df = \mathcal{L} - D_{KL}(q(f)\|p(f)) \le \mathcal{L} \tag{4}$$

More generally, $u_m$ is chosen to be a linear functional $\phi_m$ of $f$ (Lázaro-Gredilla & Figueiras-Vidal, 2009), denoted $\langle \phi_m, f \rangle \in \mathbb{C}$ with associated parameter $z_m$, in order that $u$ is Gaussian a priori. For features other than inducing points, these are termed *inter-domain* features. Let $\phi_m^*$ be such that $\langle \phi_m^*, f \rangle = \langle \phi_m, f \rangle^*$ (the complex conjugate) and let $\mathcal{K}$ be the covariance operator corresponding to $k$. That is, $[\mathcal{K}\phi_m](x_*) = \langle \phi_m^*, k(x_*, \cdot) \rangle$. Then (Bogachev, 1998, Chapter 2; Lifshits, 2012)

$$\langle \phi_m, f \rangle \sim \mathcal{N}(0, \langle \phi_m, \mathcal{K}\phi_m \rangle)$$
$$\mathbb{E}[\langle \phi_m, f \rangle \langle \phi_{m'}, f \rangle^*] = \langle \phi_m, \mathcal{K}\phi_{m'} \rangle$$
$$\mathbb{E}[f(x_*)\langle \phi_m, f \rangle] = \langle \phi_m^*, k(X_*, \cdot) \rangle$$

and for convenience define

$$c_m(x_*) = c(z_m, x_*) \overset{\text{def}}{=} \langle \phi_m^*, k(x_*, \cdot) \rangle = c^*(x_*, z_m) = \langle \phi_m, k(\cdot, x_*) \rangle \tag{5}$$

$$\bar{k}_{m,m'} = \bar{k}(z_m, z_{m'}) \overset{\text{def}}{=} \langle \phi_m, \mathcal{K}\phi_{m'} \rangle = \langle \phi_m, c(z_m, \cdot) \rangle = \langle \phi_m^*, c(\cdot, z_m) \rangle \tag{6}$$

which give the entries of the covariance matrices $K_{u\mathfrak{f}}$ and $K_{uu}$. With inducing points, $c = \bar{k} = k$. But, more generally, $p(u) = \mathcal{N}(0, K_{uu})$, and the optimal $q(u)$ is available in closed form as

$$q(u) \sim \mathcal{N}(\mu_u, \Sigma_u) \quad \text{with } \Sigma_u^{-1} = K_{uu}^{-1}(K_{uu} + \sigma^{-2} K_{u\mathfrak{f}} K_{u\mathfrak{f}}^*) K_{uu}^{-1},$$
$$\mu_u = \sigma^{-2} \Sigma_u K_{uu}^{-1} K_{u\mathfrak{f}} y \tag{7}$$

with corresponding training objective (Titsias, 2009)

$$\mathcal{F}(\mu_u, \Sigma_u) = \log \mathcal{N}(y|0, K_{u\mathfrak{f}}^* K_{uu}^{-1} K_{u\mathfrak{f}} + \sigma^2 I) - \frac{1}{2}\sigma^{-2} \text{tr}(K_{\mathfrak{f}\mathfrak{f}} - K_{u\mathfrak{f}}^* K_{uu}^{-1} K_{u\mathfrak{f}}) \tag{8}$$

wherein the structured approximation to the data precision is $A' = (K_{u\mathfrak{f}}^* K_{uu}^{-1} K_{u\mathfrak{f}} + \sigma^2 I)^{-1}$. However, by exploiting standard linear algebra results (Appendix A), the inverse and log determinant can be isolated to $B = (K_{uu} + \sigma^{-2} K_{u\mathfrak{f}} K_{u\mathfrak{f}}^*)^{-1}$ (which is the precision of the appropriately noise-corrupted features $u + K_{u\mathfrak{f}}\rho$) and $K_{uu}^{-1}$, both of which are only $M \times M$. However, in practice, the dominant cost is $O(NM^2)$ to form $K_{u\mathfrak{f}} K_{u\mathfrak{f}}^*$, since generally $M \ll N$ and the cross-covariance matrix depends nonlinearly on the hyperparameters, so must be recalculated each time Burt et al. (2020b). Put differently, the features $c_m(\cdot)$ are dependent on the hyperparameters. By choosing the linear functionals carefully, we aspire to find features which do not depend on the hyperparameters, so that $K_{u\mathfrak{f}} K_{u\mathfrak{f}}^*$ can be precomputed and stored, reducing the cost to $O(M^3)$, without compromising on feature efficiency.

---

[2]In a standard minor abuse of notation, we write the distributions over $f$ as densities, though none exist.

The posterior predictive at new points $x_*$ is calculated as

$$q(f(x_*)|x_*, x, y) = \int p(f(x_*)|x, z, u)q(u)\, du$$
$$= \mathcal{N}(f(x_*)|K_{u*}^* K_{uu}^{-1}\mu_u,\ K_{**} - K_{u*}^* K_{uu}^{-1}K_{u*} + K_{u*}^* K_{uu}^{-1}\Sigma_u K_{uu}^{-1}K_{u*}^*). \tag{9}$$

Moreover, high quality posterior samples can be efficiently generated (for example, when the number of inputs in $x_*$ is very large) by updating a random projection approximation of the prior (for example, using random Fourier features) using samples of the inducing variables (Wilson et al., 2020).

We note that the lower bound property of this training objective makes it meaningful: increases in $\mathcal{L}$ involve either increasing the marginal likelihood with respect to the hyperparameter, or reducing the KL divergence from the approximate posterior to the true posterior. This KL divergence is between the approximate and true posterior processes, giving reassurance on the quality of posterior predictive distributions (Matthews et al., 2016). Moreover, the inducing values $u$ act as a meaningful summary of the training data which can be used in downstream tasks, for example in order to make fast predictions (Wilson et al., 2020).

## 3 Related Work

There are two other main, broadly applicable, approaches to reducing the cost of learning, which are complementary:

1. using iterative methods based on fast matrix-vector multiplications (MVMs) to approximate the linear solve and log determinant, and

2. directly forming a low-rank approximation to the kernel matrix $K_{\mathfrak{ff}}$.

In the former case, the cost is reduced to $O(N^2)$ in exchange for modest error, since only a limited number of steps of the iterative methods are needed to get close to convergence in practice. This is particularly advantageous when performing operations on GPU (Gardner et al., 2018b; Pleiss et al., 2018), and when $A$ has some special structure that permits further reductions – due either to structure in the data or in $k$ (Saatçi, 2011; Cunningham et al., 2008). Direct approximations of $K_{\mathfrak{ff}}$ include by projections onto the Fourier basis for stationary kernels (Random Fourier Features, RFF (Rahimi & Recht, 2007) and variants (Lázaro-Gredilla et al., 2010; Gal & Turner, 2015), interpolating from regular grid points (stochastic kernel interpolation (Wilson & Nickisch, 2015; Gardner et al., 2018a, SKI)), or projecting onto the highest variance harmonics on compact sets (Solin & Särkkä, 2020). Notably, SKI makes use of fast MVMs with structured matrices to obtain costs which are linear in $N$ for low $D$. In contrast to the variational approach, these methods tend to approximate the posterior indirectly and the approximations may be qualitatively different to the exact posterior (see, for example, Hensman et al. (2017)). Variational methods can also be viewed as making the low rank approximation $K_{u\mathfrak{f}}^* K_{uu}^{-1}K_{u\mathfrak{f}}$ to the kernel matrix, but note that the training objective differs from simply plugging in this approximation to the marginal likelihood, as it has an additional trace term (Equation (8)). Recently, authors have attempted to incorporate nearest neighbour approximations into a variational framework (Tran et al., 2021; Wu et al., 2022).

One notable approach which does not fit into these categories is using Kalman filtering: Gaussian process regression can be viewed as solving a linear stochastic differential equation, which has $O(N)$ cost given the linear transition parameters (Särkkä et al., 2013). In practice, if the data does not have additional structure such as regularly spaced inputs, computing these transition parameters will dominate the cost.

Finally, by careful design of the prior, we can create classes of covariance function for which inference and learning are computationally cheaper (Cohen et al., 2022; Jø rgensen & Osborne, 2022). However, these are not broadly applicable in the sense that the classes of covariance function (and hence the prior assumptions) are limited, and only suitable to certain applications.

**Fourier features** If we restrict the prior to be stationary, that is $k(x, x') = k(x - x') = k(\tau)$, then $k$ has a unique spectral measure. We assume throughout that the spectral measure has a proper density

$s(\xi) = \int k(\tau)e^{-i2\pi\tau^{\top}\xi}\,d\tau$. Note that according to the convention we use here, $\int_{\mathbb{R}^D} s(\xi)\,d\xi = k(0)/(2\pi)^D$. A first attempt at hyperparameter-independent features is Fourier features, appropriately normalised by the spectral density: $\langle \phi_{1,\xi}, f \rangle = \int f(x)e^{-i2\pi x\xi}/s(\xi)\,dx$. These are independent with unbounded variance (Lázaro-Gredilla & Figueiras-Vidal, 2009; Lifshits, 2012, Chapter 3), which can be shown as follows.

$$c_1(x', \xi) = \int k(x', x)e^{-2\pi\xi^{\top}x}/s(\xi)\,dx = e^{-i2\pi\xi^{\top}x'} \tag{10}$$

$$\bar{k}_1(\xi, \xi') = \int c(x', \xi)e^{-2\pi\xi'^{\top}x'}/s(\xi')\,dx' = \int e^{-2\pi x'^{\top}(\xi-\xi')}/s(\xi')\,dx' = \delta(\xi - \xi')/s(\xi) \tag{11}$$

Here, $\delta$ is the Dirac delta. These features are unsuitable for constructing the conditional prior $p(f|u)$ – informally, the prior feature precision $K_{uu}^{-1}$ vanishes but $K_{uf}^{*}$ is finite, so the conditional prior mean $K_{fu}^{*}K_{uu}^{-1}u$ is zero (Figures 1a and 1b).

Yet the general form is promising since (i) $K_{fu}^{*}K_{fu}$ depends only on the chosen features and the training inputs $x$, so indeed if we can fix the frequencies to good values, then we can precompute this term outside of the optimisation loop, and reduce the cost of computing $A'$ to $O(M^3)$ per step (Appendix A), and also (ii) the features are independent, so $K_{uu}$ would be diagonal.

Modifications to Fourier features include applying a Gaussian window (Lázaro-Gredilla & Figueiras-Vidal, 2009) which gives finite variance but highly co-dependent features, and Variational Orthogonal Features, where $\langle \phi_m, f \rangle = \int \int e^{i2\pi\xi^{\top}x}\psi_m(\xi)/\sqrt{s(\xi)}\,d\xi f(x)\,dx$ for pairwise orthogonal $\psi_m$. This approach yields independent features, so $K_{uu}$ is diagonal, but it is challenging to find suitable sets of orthogonal functions. In both of these cases, the cross-covariance $c_m$ still depends upon the hyperparameters, and so there is usually little to no computational advantage (Burt et al., 2020a).

Variational Fourier Features (VFF) (Hensman et al., 2017) set $\langle \phi_m, f \rangle$ to a reproducing kernel Hilbert space (RKHS) inner product between the harmonics on $[a, b]$, $a < b \in \mathbb{R}$ and $f$ in 1D. Due to limiting the domain to a compact subset, Fourier transforms become discrete – that is, they become generalised Fourier series. Consequently, conditioning on only a finite subset of the frequencies works, and this gives diagonal + low-rank structure in $K_{uu}$ for lower order one-dimensional Matérn kernels. However, the covariance functions are defined on $\mathbb{R} \times \mathbb{R}$ rather than $[a, b] \times [a, b]$, so it is not straightforward to evaluate their spectra. Replacing the Euclidean inner product with an RKHS inner product permits to do this for lower order Matérn kernels, but this is not easily extended to other covariance functions, such as the spectral mixture kernel, or products of kernels. Moreover, in higher dimensions it is necessary to use a tensor product of 1D Matérn kernels, which is limiting. Hensman et al. (2017) and Dutordoir et al. (2020) note that using a regular grid of frequencies as in VFF significantly increases cost for $D > 1$ unecessarily, since features which are high frequency in every dimension are usually very unimportant. However, we demonstrate that it is possible to filter out these features (Section 4).

This approach could be generalised by replacing the Fourier basis with some other basis. Then $c_m$ is calculated using RKHS inner products with other basis functions, always yielding a hyperparameter independent $c_m$, with sparse matrices if the basis functions have compact support with little overlap. However, the need to calculate the RKHS norm of the basis functions for the elements of $K_{uu}$ limits these methods to kernels whose RKHS has a convenient explicit characterisation. In practice, this means using tensor products of 1D Matérn kernels. The recent work of Cunningham et al. (2023) is a specific example of this which uses B-splines.

Dutordoir et al (2020) used spherical harmonic features for zonal kernels on the sphere, and this can be applied to $\mathbb{R}^D$ by mapping the data onto a sphere. In this case the inducing features are well defined and independent, and this can be generalised to other compact homogeneous Riemannian manifolds. However, the harmonic expansion of $k$ on the domain must be known; for *isotropic* kernels on $\mathbb{R}^D$ restricted to the manifold, these can be computed from the spectral density (Solin & Särkkä, 2020). Yet isotropy is too limiting an assumption; one can effectively incorporate different lengthscales in each dimension by learning the mapping onto the sphere, but $\tilde{K}_{uf}$ also depends on this mapping, and so the cost returns to $O(NM^2)$.

We seek a method which can be used with a broader class of covariance functions, but retains the key computational benefits.

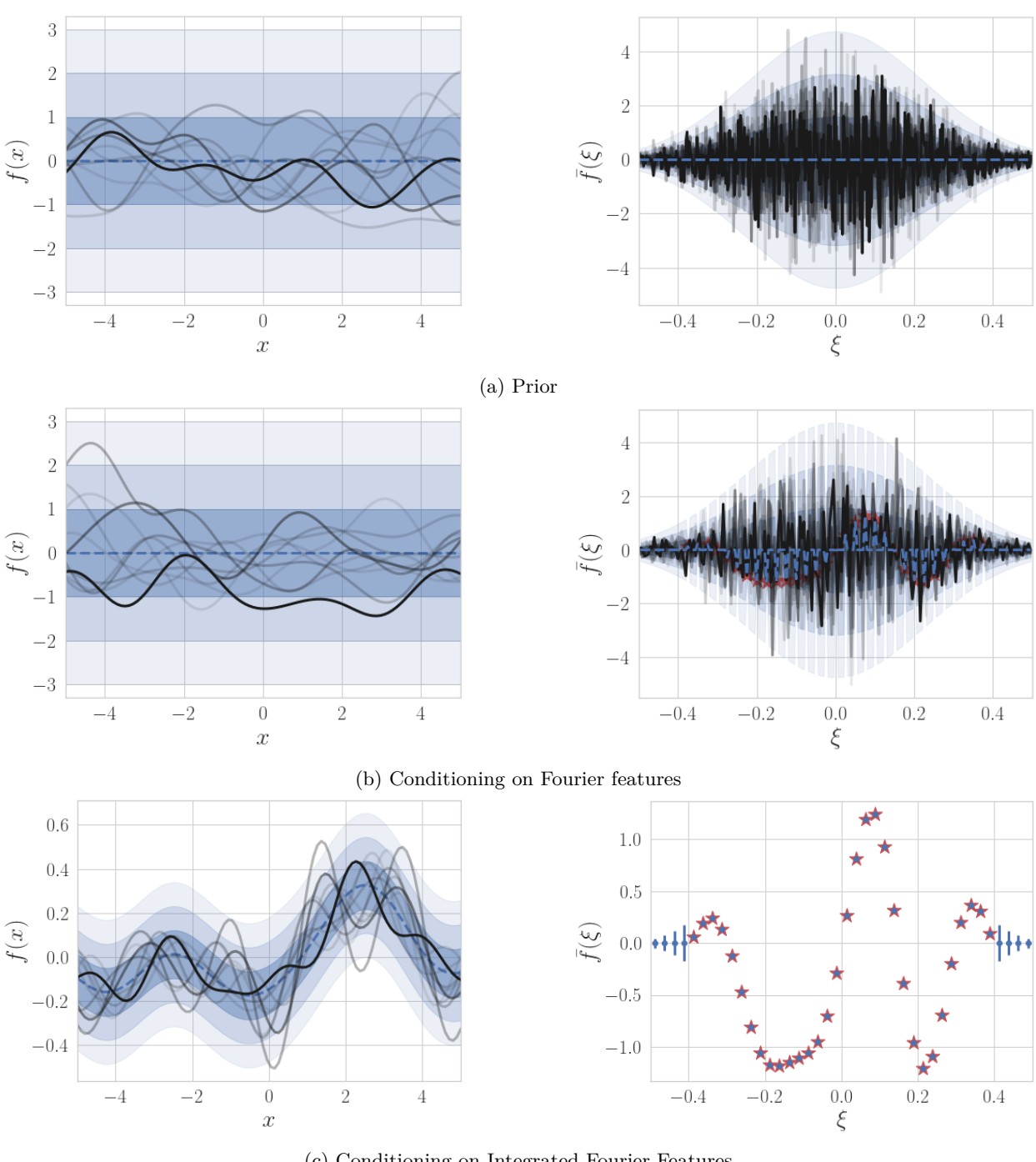

(a) Prior

(b) Conditioning on Fourier features

(c) Conditioning on Integrated Fourier Features

Figure 1: Illustration of the Integrated Fourier Feature construction. We plot the mean function (dashed), between one and three standard deviations (shaded) and sample functions in both the data and frequency domains for a squared exponential kernel with unit lengthscale. The sample functions in the data and frequency domains correspond to one another. (a) The prior's Fourier transform is white Gaussian noise whose variance is given by the spectral density. (b) We cannot condition meaningfully on come finite collection of frequencies (red stars), as this gives no information about the other frequencies – the conditional prior $p(f|u)$ in the data domain is unchanged. (c) We show only the inducing values in the frequency domain, which are averages of the surrounding region. The conditional prior is now meaningful, and the residual uncertainty is due to high frequency content not included in the features.

**Choosing** $z$   It is well known that optimising inducing inputs is usually not worth the extra computational cost compared to a good initialisation. We briefly review the initialisation methods for different features described above.

For inducing points, Burt et al. (2020b) show that sampling from a $k$-DPP (determinental point process; with $k = M$, and the kernel used in the DPP is the same as the GP prior's) performs well, both in theory and in practice. Since that initialisation is hyperparameter-dependent, they alternate between optimising the hyperparameters and sampling $z$ in a variational expectation maximisation (EM) approach.

For VFF as described by Hensman et al. (2017), a rectangular grid of regularly spaced frequencies must be used, which they select (optimally in 1D) to be centred around the origin. In higher dimensions, the regular grid leads to including suboptimal frequencies in the corners of the grid. A construction which leads to a more feature efficient set of frequencies is the following. Create a rectangular grid, and then discard the features which are not within a given ellipsoid, where the ellipsoid's axes should be chosen to the proportional to the bandwidth of the spectral density in that dimension (which is inversely proportional to the lengthscale). This corresponds to discarding the corresponding rows in $K_{u\mathfrak{f}}$, and the corresponding row and column in $K_{uu}$.

For spherical harmonics, the optimal choice is to use the frequencies which have the highest variance. For many covariance functions (for example, those constructed from monotonically decreasing stationary kernels on $\mathbb{R}^D$ using the method of Solin & Särkkä (2020)) this corresponds to choosing the first $M$ frequencies.

For B-spline features, a grid of regularly spaced basis functions which covers a rectangular domain containing the data is used, and the sparsity of the matrices is used to make the method efficient – features which are not strongly correlated with the data also contribute less to the computational cost.

## 4   Integrated Fourier Features

We are not able, in general, to integrate out the Dirac delta in Equation (11) and retain the desirable computational properties without introducing further approximations. We propose to average Fourier features over *disjoint* intervals of width $\varepsilon$ (Figure 1c), and approximate the spectral density as constant over the integration width. We focus on $D = 1$ to lighten the presentation here; to enforce that the intervals are disjoint we require $|z_m - z_{m'}| \geq \varepsilon$ for any $m \neq m'$.

$$\langle \phi_{2,m}, f \rangle \overset{\text{def}}{=} \varepsilon^{-1} \int_{z_m - \varepsilon/2}^{z_m + \varepsilon/2} \int f(x) e^{-i2\pi\xi x} \, dx/s(\xi) \, d\xi = \varepsilon^{-1} \int_{z_m - \varepsilon/2}^{z_m + \varepsilon/2} \langle \phi_{1,\xi}, f \rangle \, d\xi$$

$$c_2(x', z_m) = \varepsilon^{-1} \int_{z_m - \varepsilon/2}^{z_m + \varepsilon/2} c_1(x', \xi) \, d\xi \approx e^{-i2\pi z_m x'} \tag{12}$$

$$\bar{k}_2(z_m, z_{m'}) = \varepsilon^{-2} \int_{z_m - \varepsilon/2}^{z_m + \varepsilon/2} \int_{z_{m'} - \varepsilon/2}^{z_{m'} + \varepsilon/2} \bar{k}_1(\xi, \xi') \, d\xi' \, d\xi \approx \varepsilon^{-1} \delta_{m-m'}/s(z_m) \tag{13}$$

Now, $\delta$ is the Kronecker delta. Note that if the intervals were not chosen to be disjoint then only the last line would change. The advantage of choosing disjoint intervals is to make $K_{uu}$ diagonal, which will simplify later analysis, as well as modestly reducing the computational cost. But recall that the inversion and log determinant of $B$ is still required, and this matrix remains dense.

### 4.1   Convergence

Now, if we calculate covariance matrices using the numerical approximation detailed above, and use this to evaluate the collapsed variational objective of Equation (8), we no longer have a proper variational objective in the sense of Equation (8). In order to distinguish the proper objective from our approximation, we introduce the notation $\mathfrak{F}$ for the approximate objective.

In this section we show that the approximation $\mathfrak{F}$ converges to $\mathcal{L}$ at a reasonable rate as $M \to \infty$, for a broad class of covariance functions, showing that these features are efficient and produce good approximations for

the purposes of hyperparameter optimisation. Since we no longer have the interpretation of reducing the KL between posterior processes, we provide additional results to give reassurance about the quality of the approximate posterior predictive distribution.

Firstly, we subtly transform the features to simplify the analysis. Note that applying an invertible linear transformation $T$ to the features has no impact on inference or learning. That is, if we transform $u'$ with mean and covariance $\mu_{u'}, \Sigma_{u'}$ to $u = Tu'$, then $K_{u\mathfrak{f}} = \mathbb{E}[uf(x)^\top] = \mathbb{E}[Tu'f(x)^\top] = TK_{u'\mathfrak{f}}$, and similarly $K_{uu} = TK_{u'u'}T^*$. Then from Equation (7) it follows that if we optimise after transforming, the optimal $\Sigma_u^{-1} = T^{-*}\Sigma_{u'}^{-1}T^{-1}$ and the optimal $\mu_u = T\mu_{u'}$, as would be expected from optimising before transforming. Furthermore the collapsed objective of Equation (8) and posterior predictive mean and covariance of Equation (9) are left unchanged.

For the analysis, instead of normalising by the spectral density, we normalise by its square root. This has the advantage that we do not need any approximation for $\bar{k}$, only for $c$. This simplifies later calculations.

$$\langle \phi_{3,m}, f \rangle \stackrel{\text{def}}{=} \varepsilon^{-1} \int_{z_m - \varepsilon/2}^{z_m + \varepsilon/2} \int f(x)e^{-i2\pi\xi x}\, dx / \sqrt{s(\xi)}\, d\xi = \varepsilon^{-1} \int_{z_m - \varepsilon/2}^{z_m + \varepsilon/2} \langle \phi_{1,\xi}, f \rangle \sqrt{s(\xi)}\, d\xi$$

$$c_m(x') = c_3(x', z_m) = \varepsilon^{-1} \int_{z_m - \varepsilon/2}^{z_m + \varepsilon/2} c_1(x', \xi)\sqrt{s(\xi)}\, d\xi \approx \sqrt{s(z_m)}e^{-i2\pi z_m x'} = \hat{c}_m(x')$$

$$\bar{k}_{m,m'} = \bar{k}_3(z_m, z_{m'}) = \varepsilon^{-2} \int_{z_m - \varepsilon/2}^{z_m + \varepsilon/2} \int_{z_{m'} - \varepsilon/2}^{z_{m'} + \varepsilon/2} \bar{k}_1(\xi, \xi')s(\xi)\, d\xi'\, d\xi = \varepsilon^{-1}\delta_{m-m'}$$

We proceed without the subscript 3 hereafter for brevity. The new approximate features are a straightforward invertible linear transformation of the previous features (in particular, $\hat{u}_m = \hat{u}_{3,m} = \sqrt{s(z_m)}\hat{u}_{2,m}$). We analyse how well $\hat{u}$ approximates $u$, but in practice we use a real-valued version of the equivalent approximate features $\hat{u}_2$ in order to have $\hat{c}$ independent of the hyperparameters (Appendix B). We defer the details of the proofs, particularly for higher dimensional inputs, to Appendix C.

For convergence of the objectives, we use the following result of Burt et al. (2019).

$$\mathbb{E}_y[D_{KL}(q(f) \,\|\, p(f|y))] = \mathbb{E}_y[\mathcal{L} - \mathcal{F}] \leq \frac{t}{\sigma^2} \tag{14}$$

$$t = \text{tr}(K_{\mathfrak{f}\mathfrak{f}} - \underbrace{K_{u\mathfrak{f}}^* K_{uu}^{-1} K_{u\mathfrak{f}}}_{Q_{\mathfrak{f}\mathfrak{f}}}) \tag{15}$$

where $y$ is distributed according to Equation (1). The first equality follows for any well-defined inducing features (Matthews et al., 2016)–that is, those where the inducing features can be a priori described as a linear functional of the prior process. We now detail the additional assumptions.

**A1** Let $\tilde{s} = s/(v\sigma^2)$ be the normalised spectral density, which is assumed to exist, with $v = k(x, x)/\sigma^2$. We assume that the normalised spectral density has a tail bound

$$\int_\rho^\infty \tilde{s}(\xi)\, d\xi \leq \beta\rho^{-q} \tag{16}$$

for any $\rho > 0$ and some $\beta, q > 0$.

**A2** The second derivative of $s$ is bounded.

**A3** The first derivative has a relative bound

$$\left| \frac{ds(\xi)}{d\xi} \right| \leq 2Ls(\xi) \implies \left| \frac{d\sqrt{s(\xi)}}{d\xi} \right| \leq L\sqrt{s(\xi)} \tag{17}$$

for some $L > 0$, where the implication follows wherever $s(\xi) > 0$. For example, this is satisfied if $s$ and its first derivative are bounded everywhere, which is the case for widely used covariance functions.

**A4** The frequencies are chosen according to the regular grid $z_m = (m - 1/2)\varepsilon - M/2$ for $M$ even. This requirement can be relaxed in practice. The key requirement is that $\bigcup_m [z_m - \varepsilon/2, z_m + \varepsilon/2] \to \mathbb{R}$; $\varepsilon$ could also be varied as a function of $m$, with the convergence rate dominated by the largest.

The higher dimensional generalisation of these are the standard assumptions. We use the following simple result, proved in the supplement.

**Lemma 4.1.** *Under assumptiona A3 and A4,*

$$c_m(x) = \hat{c}_m(x)(1 + O(\varepsilon))$$

**Theorem 4.2.** *For $y$ sampled according to Equation* (1), *under assumptions A1 to A4, for any $\Delta, \delta > 0$ there exists $M_0, \alpha_0 > 0$ such that*

$$\mathbb{P}\left[\frac{\mathcal{L} - \mathfrak{F}}{N} \geq \frac{\Delta}{N}\right] \leq \delta$$

*for all $M \geq M_0$, and with*

$$M \leq \left(\frac{\alpha_0}{\Delta\delta}N\right)^{\frac{q+3}{2q}}$$

*Moreover, there exists $M_1, \alpha_1 > 0$ such that for all $M > M_1$,*

$$\mathbb{P}\left[\frac{\mathcal{L} - \mathcal{F}}{N} \geq \frac{\Delta}{N}\right] \leq \delta$$

*for all $M \geq M_1$, and with*

$$M \leq \left(\frac{\alpha_1}{\Delta\delta}N\right)^{\frac{q+3}{2q}}$$

*Proof.* We sketch the 1D case here. Let $\hat{t} = \mathrm{tr}(K_{\mathfrak{ff}} - \hat{K}_{u\mathfrak{f}}^* K_{uu}^{-1} \hat{K}_{u\mathfrak{f}}) = \mathrm{tr}(K_{\mathfrak{ff}} - \hat{Q}_{\mathfrak{ff}})$. First we show that $\hat{t}/N\sigma^2 \in O(M^{-2q/(q+3)})$.

$$\frac{\hat{t}}{N\sigma^2} = \frac{1}{N\sigma^2} \sum_n \left(\underbrace{k(x_n, x_n)}_{v\sigma^2} - \varepsilon \sum_m \hat{c}_m(x_n)\hat{c}_m^*(x_n)\right) \tag{18}$$

$$= v\left(1 - \varepsilon \sum_m \tilde{s}(z_m)\right) \tag{19}$$

$$= v\left(1 - \int_{-(M+1)\varepsilon/2}^{(M+1)\varepsilon/2} \tilde{s}(\xi)\,d\xi\right) + vE_1 \tag{20}$$

$$= 2v\int_{M\varepsilon/2}^{\infty} \tilde{s}(\xi)\,d\xi + vE_1 \tag{21}$$

where $E_1 = \int_{-(M+1)\varepsilon/2}^{(M+1)\varepsilon/2} \tilde{s}(\xi) - \varepsilon \sum_m \tilde{s}(z_m) + O(M\varepsilon^3)$. The integral term in the last line is in $O((M\varepsilon)^{-q})$ by assumption A1, and $E_1 \in O(M\varepsilon^3)$ from standard bounds on the error of the midpoint approximation (using A2).

We must have that $\varepsilon \to 0$ as $M \to \infty$ to make the midpoint approximation exact, yet we must have $M\varepsilon \to \infty$ to ensure the features cover all frequencies. By optimising the trade-off, we get the stated bound.

To complete the proof, for $t$ we could immediately apply Equation (14). But for $\hat{t}$, we adapt the result of Equation (14) to show $\mathbb{E}_y[|\mathcal{L} - \mathfrak{F}|/N] \leq \hat{t}/N\sigma^2$ for sufficiently large $M$, using Lemma 4.1 (assumptiona A3, A4). By applying Markov's inequality, we complete the proof. $\square$

*Remark* 4.3. The case of subgaussian spectral densities (such as for the squared exponential covariance function) is $q \to \infty$, which yields $M \in O(\sqrt{N})$. Note that this is due to the $O(M\varepsilon^3)$ terms which arise due to approximating the spectral density as constant. Intuitively, it appears that if there were no numerical

approximation, the cost would be dominated by the amount of spectrum in the tails, such that convergence for subgaussian tails would be possible with $M \in O(\log N)$ for sufficiently small $\varepsilon$, as with inducing points or eigenfunctions with the squared exponential covariance function Burt et al. (2019).

Though this demonstrates that the features are suitable for learning, we may wish to use the same features in making predictions. We no longer can use the bound in the process KL, but we show that the posterior predictive marginals converge with comparable or better rate than the objective for any choice of variational distribution.

**Theorem 4.4.** *For the optimised $\mu_u, \Sigma_u$ (to maximise $\mathfrak{F}$), let the posterior predictive at any test point $x_*$ using the exact features $u$ have mean and variance $\mu, \Sigma$, and with the approximate features $\hat{u}$ have mean and variance $\hat{\mu}, \hat{\Sigma}$. Then, under assumptions A3 and A4,*

$$|\mu - \hat{\mu}| \in O(M\varepsilon^2)$$
$$|\Sigma - \hat{\Sigma}| \in O(M^2\varepsilon^3)$$

*In particular, allowing $\varepsilon \sim M^{-\frac{q+1}{(q+3)}}$ as in the proof of Theorem 4.2 (see Appendix C), we have*

$$|\mu - \hat{\mu}| \in O\left(M^{-\frac{q-1}{q+3}}\right) \tag{22}$$

$$|\Sigma - \hat{\Sigma}| \in O\left(M^{-\frac{q-3}{q+3}}\right) \tag{23}$$

*Proof.* Use the definitions in Equation (7) and apply Lemma 4.1 and the triangle equality. $\square$

## 4.2 Higher dimensions

In higher dimensions, we modify the assumptions as follows.

**A1** Assume that $\tilde{k}$'s spectral measure admits a density, and denote this by $\tilde{s}$ and assume that the density admits a tail bound

$$\int_\rho^\infty \cdots \int_\rho^\infty \tilde{s}(\xi)\, d\xi_1 \ldots d\xi_D \le \beta \rho^{-qD} \tag{24}$$

for any $\rho > 0$ and some $\beta, q > 0$.

**A2** The spectral density's second derivative is bounded.

**A3** The spectral density's first derivatives are bounded as

$$\frac{\partial s(\xi)}{\partial \xi_d} \le 2Ls(\xi) \implies \frac{\partial \sqrt{s(\xi)}}{\partial \xi_d} \le L\sqrt{s(\xi)} \tag{25}$$

for some $L > 0$ (where the second expression follows wherever $s(\xi) > 0$). For example, this would be satisfied if the spectral density and its first derivative are both bounded everywhere, which includes widely used covariance functions.

**A4** Let the inducing frequencies be an analogous regular grid in higher dimensions, with $M^{1/D} \in \mathbb{Z}$ even. That is, for a multi-index $m_{1:D}$,

$$[z_{m_{1:D}}]_d = (-(M^{1/D}+1)/2 + m_d)\varepsilon \quad \text{for } d \in \{1:D\}. \tag{26}$$

Then the main result (Theorem 4.2) and the predictive bounds in terms of $q$ in Theorem 4.4 are unchanged. See Appendix C for details.

Assumption A4 requires a full regular grid of features, which means the number of features increases exponentially in $D$. In common with previous work (Hensman et al., 2017; Cunningham et al., 2023) this can

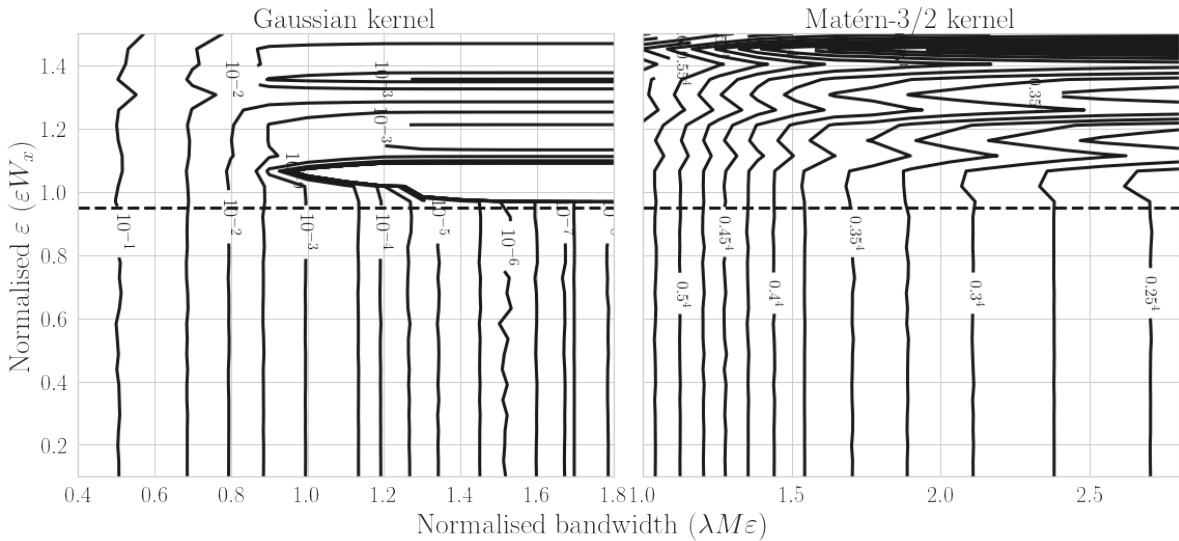

Figure 2: Gap between the log marginal likelihood and the training objective ($\mathcal{L} - \mathfrak{F}$) for different settings of $M$, $\varepsilon$ for data sampled from a GP with a Gaussian (left) or Matérn-3/2 (left) kernel. In each case the hyperparameters are set to their groundtruth values, where the lengthscale is $\lambda$. The inputs are samples from a uniform distribution centred on 0 and with width $W_x$. The horizontal line is at 0.95.

be avoided if the covariance function can be assumed to be additive over dimensions. Otherwise, we can improve the computational cost by using a subset of the grid points where the spectral density is highest. Although this improves the cost in higher dimensions, the scaling is in general still exponential in $D$. This limits IFF's applicability to lower dimensional spatial or spatiotemporal modelling applications.

### 4.3 Choosing the approximation parameters

**Choosing $M$**  These need to be selected whenever using SGPR. In IFF, the faster $s$ decays, the lower $M$ we should need for a good approximation to the log marginal likelihood (Theorem 4.2) and Figure 2 shows that $M$ need not be very large (in each dimension) to get good performance; we need $M\varepsilon$ at around the approximate bandwidth of the covariance function, which increases as the lengthscale $\lambda$ reduces. If we know a priori how small the lengthscale might become – for example by examining the Fourier transform of the data, from prior knowledge, or by training a model on small random subsets of the data – then we can use this to select $M$. Note that if the lengthscale is shorter, we would expect $M$ to need to be larger for SGPR also, as we would need inducing points to be placed closer together. In practice, as with all SGPR methods, $M$ controls the trade-off between computational resources and performance, and can be set as large as needed to get satisfactory performance for the applicaiton, or as large as the available resources allow.

**Choosing $z$**  Given a fixed $\varepsilon, M$, the optimal choice is to choose frequencies spaced by $\varepsilon$ in the regions of highest spectral density, comparable to spherical harmonics. For monotonically decreasing spectral densities maximised at the origin, such as the Gaussian or Matérn kernels, this corresponds exactly to our refined construction for VFF in Section 3, which involved choosing grid points which were contained within an ellipsoid whose axes were inversely proportional to the lengthscale in each dimension. To avoid dependence on the hyperparameters (which would undo the benefits of being able to precompute $A'$), we opt in practice for a spherical threshold.

**Choosing $\varepsilon$**  When adding more features, we can either cover a higher proportion of the prior spectral density or reduce $\varepsilon$. In the full proof of Theorem 4.2 (Appendix C), as the spectral density gets heavier tailed, $\varepsilon$ approaches $O(M^{-1})$. Then $M\varepsilon$ is almost constant, which suggests this part tends to dominate. That is, once $M\varepsilon$ is sufficiently large, we should add more features by reducing $\varepsilon$ rather than increasing

$M$. However, we explore the trade-off between increasing $M\varepsilon$ and decreasing $\varepsilon$ numerically (Figure 2) by plotting the gap between the IFF bound and the log marginal likelihood, varying the bandwidth covered and the size of $\varepsilon$ relative to the inverse of the data width ($W_x \approx \max_n x_n - \min_n x_n$). The model and data generating process use a kernel with lengthscale $\lambda$. We see that in practice, as long as $\varepsilon^{-1}$ is below the inverse of the data width, the gap is not very sensitive to its value. Thus for the experiments, we conservatively set $\varepsilon_d = 0.95/(\max_{n,d} x_{nd} - \min_{n,d} x_{nd})$.

This result appears to be in tension with the theory. However, we note that in the proof we are effectively interested in producing a good approximation of the function across the whole domain (we make no assumptions about the input locations). But in Figure 2, we explicitly take into account where the data is. Intuitively, our construction involves approximating the function with regularly sampled frequencies. Then it should be possible to construct a good approximation to a function on an interval of width $W_x$ as long as the sample spacing is no more than $1/W_x$, as a Fourier dual to the classical Nyquist-Shannon sampling theorem (see, for example, Chapter 5, Vetterli et al., 2012)

**Other covariance functions**  We have so far assumed that the spectral density is available in closed form. However, we only need regularly spaced point evaluations of the spectral density, for which it suffices to evaluate the discrete Fourier transform of regularly spaced evaluations of the covariance function. This adds, at worst, $O(M^2)$ computation to each step.

### 4.4  Limitations

IFF can be used for faster learning for large datasets in low dimensions, which matches our target applications. Typically, it will perform poorly for $D \gtrsim 4$, and both in this case and for low $N$, we expect SGPR to outperform all alternatives, including IFF, and our analysis and evaluation are limited to the conjugate setting.

IFF is limited to stationary priors; while these are the most commonly used, they are not appropriate for many spatial regression tasks of interest, and a fast method for meaningful non-stationary priors would be a beneficial extension. Amongst those stationary priors, we require that the spectral density is sufficiently regular; this is satisfied for many commonly used covariance functions, but not for periodic covariance functions, where the spectral measure is discrete. While IFF can be used with popular covariance functions for modelling quasi-periodic functions (such as a product of squared exponential and periodic covariance functions, or the spectral mixture kernel) if the data has strong periodic components, the maximum marginal likelihood parameters will cause the spectral density to collapse towards a discrete measure (for example, learning very large lengthscales).

Finally, we have left some small gaps between theory and practice, both in how to select the tunable parameter $\varepsilon$, and in characterising the quality of the posterior predictive distribution.

## 5  Experiments

We seek to show that IFF gives a significant speedup for large datasets in low dimensions, with a particular focus on spatial modelling. Amongst other fast sparse methods, we compare against VFF and B-Spline features. For spherical harmonics, learning independent lengthscales for each dimension is incompatible with precomputation. In any case, we found that we were unable to successfully learn reasonable hyperparameters with that method in our setting, except if the number of feature was very small. For a conventional (no pre-compute) sparse baseline, we use inducing points sampled according to the scheme of Burt et al. (2020a). For our synthetic experiments, we also used inducing points initialised using $k$-means and kept fixed. For the real-world spatial datasets, we also tested SKI, due to its reputation for fast performance, and its fairly robust implementation.

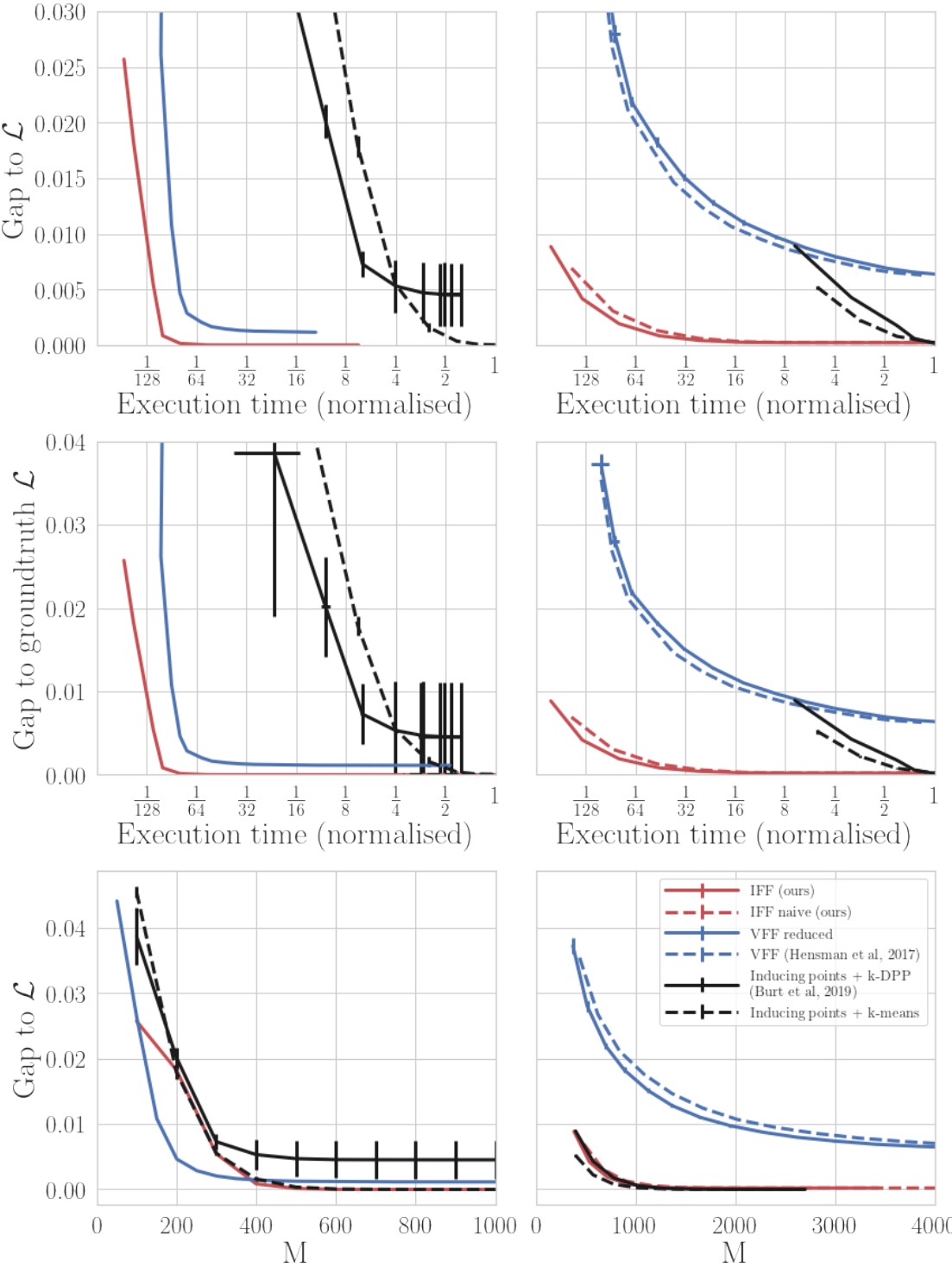

Figure 3: Comparing standard sparse Gaussian process regression (black) to IFF (red) for data generated from a prior with Gaussian covariance function in 1D (left) and 2D (right). Lower and the to the left is better. The groundtruth $\mathcal{L}$ is $\mathcal{L}$ evaluated at the groundtruth hyperparameters, whereas in the other rows, $\mathcal{L}$ is evaluated at the learnt hyperparameters. The gaps are normalised by $N$, and execution time is normalised by the longest. The bottom row shows feature efficiency, whereas the upper rows show computational efficiency.

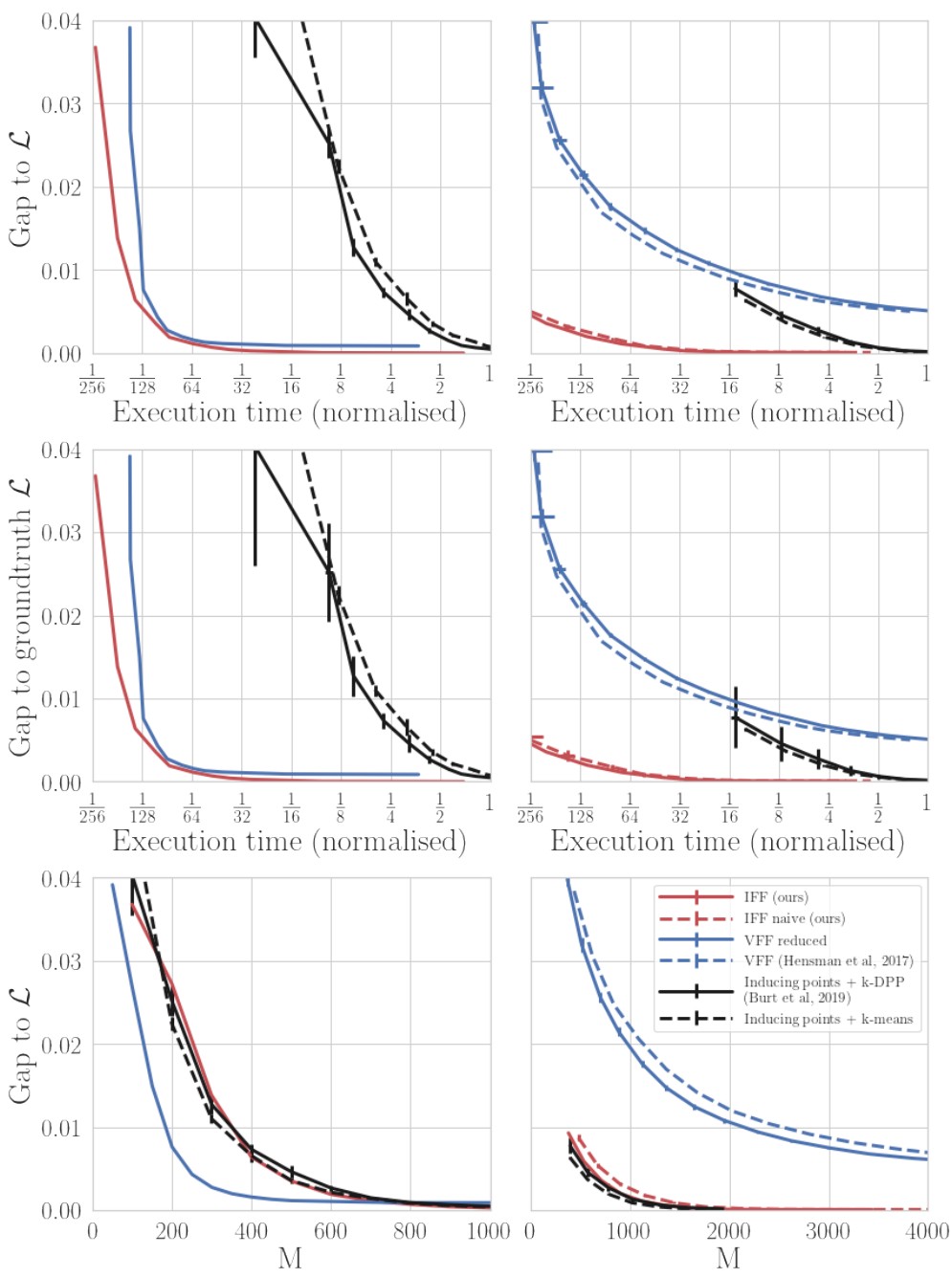

Figure 4: As Figure 3, but with the data sampled from a Matérn-5/2 GP. The picture is broadly comparable, but VFF now more closely matches the prior, so the drop in feature efficiency is far less in higher dimensions.

## 5.1 Synthetic datasets

First we consider a synthetic setting where the assumptions of Theorem 4.2 hold. We sample from a GP with a Gaussian covariance function, and compare the speed of variational methods in 1 and 2 dimensions. We use a small ($N = 10\,000$) dataset in order that we can easily evaluate the log marginal likelihood at the learnt hyperparameters. Where possible, we use the same (squared exponential) model for learning; for VFF, we use a Matérn-5/2 kernel in 1D, and a tensor product of Matérn-5/2 covariance functions, since this is the best approximation to a Gaussian kernel which is supported. Further details are in Appendix D.

Additionally, in the 2D setting, we use both the naive set of features (a regular, rectangular grid), and the refined set of features described in Section 3.

IFF generally has slightly lower gap to the marginal likelihood at the learnt optimum for any $M$ than other fast variational methods (Figure 3, bottom row), but because the $O(NM^2)$ work is done only once, it and the other fast sparse methods are much faster to run than inducing points (Figure 3, top two rows). Note the logarithmic time scale on the plots: for a specified threshold on the gap to the marginal likelihood, IFF is often around 30 times faster than using inducing points.

The experiments demonstrate the issues with the limited choice of prior with methods such as VFF. In 1D, the Matérn-5/2 kernel is a good approximation to Gaussian, so the performance is similar to IFF. But for 2D, the product model is a much worse approximation of the groundtruth model. We expect to see a similar pattern for B-spline features, but we were unable to run the method in our synthetic setting due to unresolved issues in the implementation of Cunningham et al. (2023) which did not arise in the real-world experiments. When we reproduce the same experiment, but with data sampled from a Matérn-5/2 GP, the results are similar, but with a much smaller gap between VFF and the other methods in the 2D case (Figure 4).

We note that in the 1D setting, when the data is sampled from a prior with Gaussian covariance function, the $k$-DPP method gets stuck at a local optimum, which is a known risk with that method. We did not find this to be an issue in any of the other experiments.

The refined feature set for higher dimensional IFF and VFF (solid lines) is indeed slightly more feature efficient than the naive approach (dashed lines; Figure 3, bottom right panel). But in practice, we find that the computational overhead to selecting better features outweighs the savings from using fewer features for VFF, thought not for IFF (Figure 3, upper and middle right panels).

## 5.2 Real World Datasets

We now compare training objective and test performance on three real-world spatial modelling datasets of increasing size, and of practical interest. We plot the root mean squared error (RMSE) and negative log predictive density (NLPD) on the test set along with the training objective and run time in Figures 5 and 6 using five uniformly random 80/20 train/test splits. For inducing points, we always use the method of Burt et al. (2020b). The time plotted is normalised per split against inducing points. Further training and dataset details are in Appendix D.

We add SKI to the comparison here, since it is the most widely used alternative to variational methods in this setting. For the variational methods, both the time and performance are implicitly controlled by the number of features $M$; as $M$ is increased the performance improves and the time increases, that is we move along the curve to the right. This is a very useful property, since we can select $M$ according to our available computational budget and be fairly confident of maximising performance. With SKI, the equivalent parameter is the grid size, and similarly increasing the grid size generally improves performance. However, when the grid size is low, optimisation can take longer, so for example for the temperature dataset, we move to the left as the grid size is increased (Figure 6; top row). For the other datasets, we only plot the SKI with the grid size automatically selected by the reference implementation.

SKI generally has very good predictive means, leading to low RMSE, and is very fast, but the predictive variances are poor, generally being too low and producing some negative values. Ignoring the negative

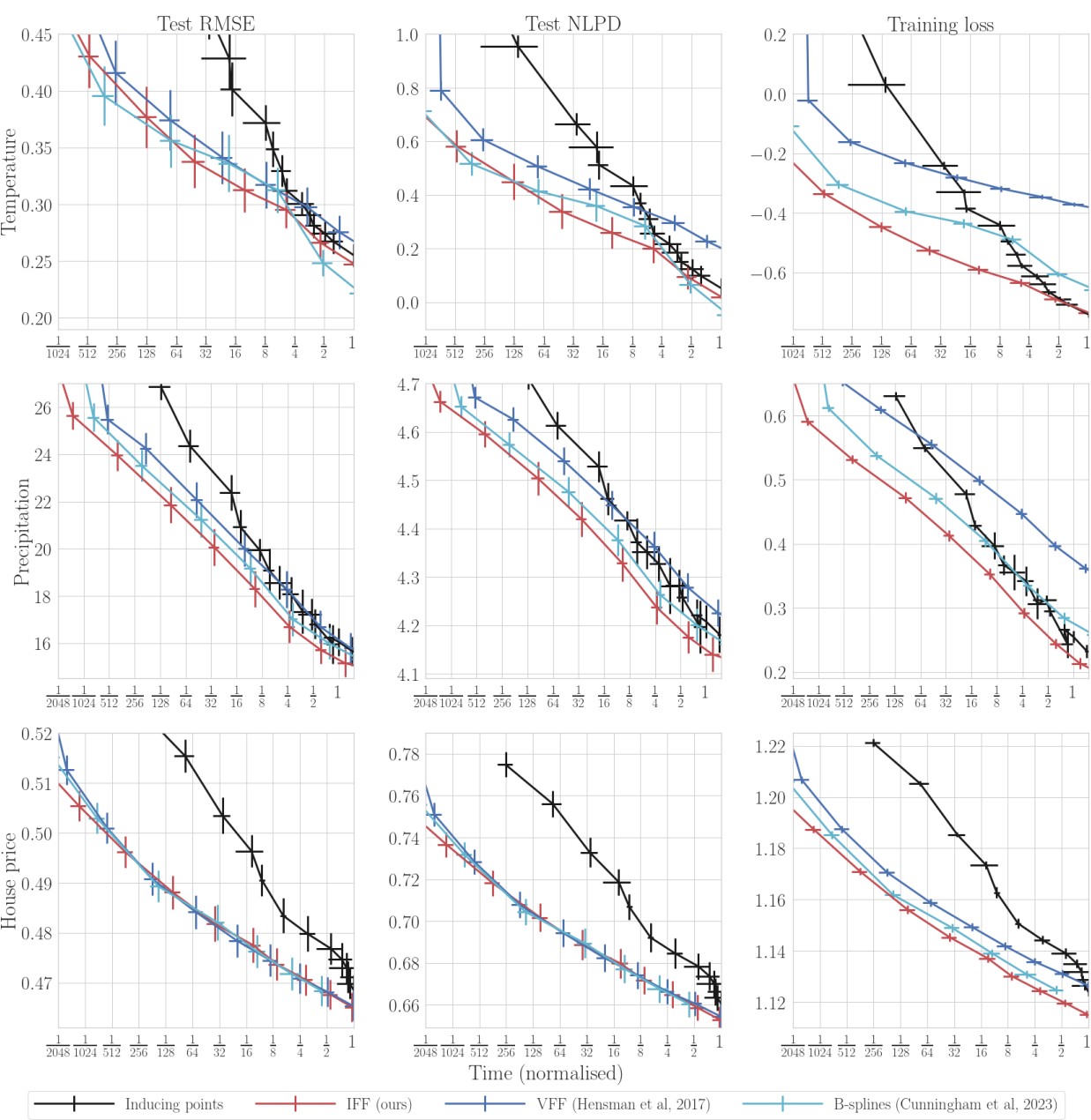

Figure 5: Performance curves for real world datasets of increasing size (the top row is the smallest). Lower and to the left is better.

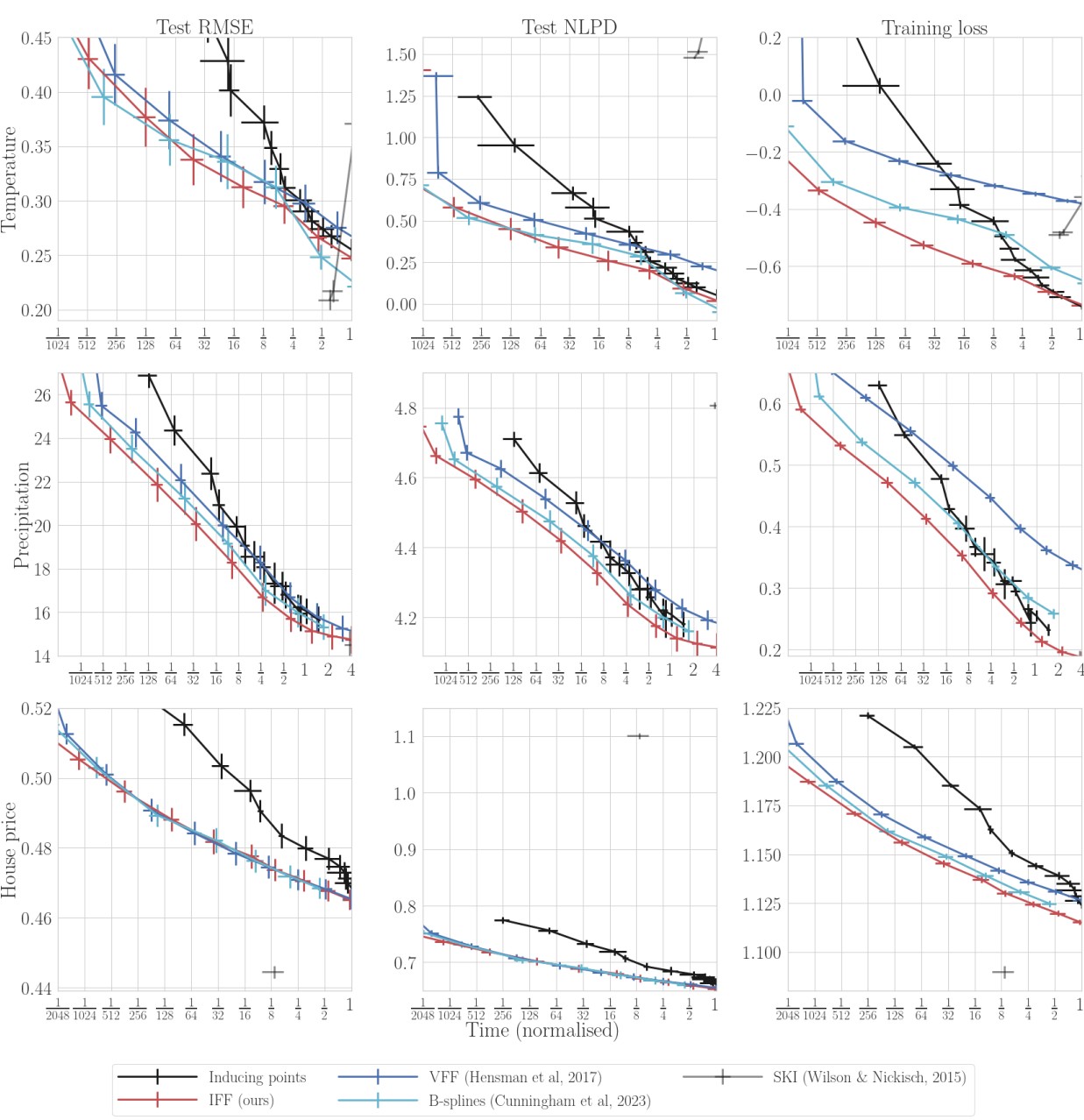

Figure 6: As Figure 5, but with SKI included. SKI exhibits very favourable and fast predictive mean performance, but its predictive variances are poor, leading to very large NLPD.

values, the NLPD is much worse for SKI than for the variational methods, which is highly undesirable in a probabilistic method.

We exclude SKI in Figure 5 in order to zoom in on the curves for the variational methods. We are interested in the regime where $M \ll N$; as we move to the right and $M$ is similar to $N$, inducing points will become competitive with the faster methods, since the $O(M^3)$ cost dominates. Comparing IFF to VFF, we see that always performs at least as well, and produces a substantially better performance for a given time on the temperature and precipitation datasets, due to a more flexible choice of covariance function – in particular, note that the training objective of IFF is substantially lower than VFF on these datasets, but comparable to that of inducing points, which uses the same covariance function.

The B-spline features are also limited in choice of covariance function, but the sparse structure of the covariance matrix leads to a much better performance. Nonetheless, despite involving a dense matrix inverse, IFF has a significant advantage on the precipitation dataset (around 25-30% faster on average), and is comparable on the houseprice dataset.

Compared to the idealised, synthetic, setting, inducing points become very competetive in the higher resource setting towards the right hand side of each plot, particularly for the smallest dataset (temperature). But for more performance thresholds, we find that fast variational methods offer a substantial improvement, with typically more than a factor of two speedup.

## 6    Conclusions

Integrated Fourier features offer a promising method for fast Gaussian process regression for large datasets. There are significant cost savings since the $O(N)$ part of the computation can be done outside of the loop, yet they support a broad class of stationary priors. Crucially, they are also much easier to analyse than previous work, allowing for convergence guarantees and clear insight into how to choose parameters.

They are immediately applicable to challenging spatial regression tasks, but a significant limitation is the need to increase $M$ exponentially in $D$. Further methods to exploit structure in data for spatiotemporal modelling tasks ($D = 3$ or 4) is an important line of further work. More broadly, an interesting direction is to consider alternatives to the Fourier basis which can achieve similar results for non-stationary covariance functions, which are crucial for achieving state of the art performance in many applications. Finally, a worthwhile direction for increased practical use would be to develop suitable features to quasi-periodic priors such as those arising from the spectral mixture kernel, or a product of periodic and Gaussian kernels.

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

## A  Computation

Recall the collapsed objective.

$$\mathcal{F}(\mu_u, \Sigma_u) = \log \mathcal{N}(y|0,\ K_{u\mathfrak{f}}^* K_{uu}^{-1} K_{u\mathfrak{f}} + \sigma^2 I) - \frac{1}{2}\sigma^{-2} \mathrm{tr}(K_{\mathfrak{f}\mathfrak{f}} - K_{u\mathfrak{f}}^* K_{uu}^{-1} K_{u\mathfrak{f}}) \tag{27}$$

$$= -\frac{1}{2}\log|K_{u\mathfrak{f}}^* K_{uu}^{-1} K_{u\mathfrak{f}} + \sigma^2 I| - \frac{1}{2}y^\top (K_{u\mathfrak{f}}^* K_{uu}^{-1} K_{u\mathfrak{f}} + \sigma^2 I)^{-1}y - \frac{1}{2}\sigma^{-2}\mathrm{tr}(K_{\mathfrak{f}\mathfrak{f}} - K_{u\mathfrak{f}}^* K_{uu}^{-1} K_{u\mathfrak{f}}) \tag{28}$$

With $\bar{y} = K_{u\mathfrak{f}}y$, $\sum_n y_n^2 = \nu^2$, we apply the Woodbury identity to the inverse in the quadratic form.

$$(K_{u\mathfrak{f}}^* K_{uu}^{-1} K_{u\mathfrak{f}} + \sigma^2 I)^{-1} = \sigma^{-2}I - \sigma^{-4}K_{u\mathfrak{f}}^*(K_{uu} + \sigma^{-2}K_{u\mathfrak{f}}K_{u\mathfrak{f}}^*)^{-1}K_{u\mathfrak{f}}$$

$$\implies y^\top(K_{u\mathfrak{f}}^* K_{uu}^{-1} K_{u\mathfrak{f}} + \sigma^2 I)^{-1}y = \sigma^{-2}\nu^2 - \sigma^{-4}\bar{y}^\top(K_{uu} + \sigma^{-2}K_{u\mathfrak{f}}K_{u\mathfrak{f}}^*)^{-1}\bar{y}$$

For the log determinant, we can use the matrix determinant lemma.

$$|\sigma^2 I + K_{u\mathfrak{f}}^* K_{uu}^{-1} K_{u\mathfrak{f}}| = |K_{uu} + \sigma^{-2}K_{u\mathfrak{f}}^* K_{u\mathfrak{f}}||K_{uu}|^{-1}|\sigma^2 I_n|$$

Finally, we write down the trace directly using the fact that $K_{uu}$ is diagonal.

$$\mathrm{tr}(K_{\mathfrak{f}\mathfrak{f}} - K_{u\mathfrak{f}}^* K_{uu}^{-1} K_{u\mathfrak{f}}) = \sum_n \left( k(x_n, x_n) - \sum_m |c_m(x_n)|^2 / \bar{k}_{mm} \right)$$

We can combine the above to get an easy to evaluate expression for $\mathcal{F}$, and replacing the inter-domain cross covariance matrices with their numerical approximations, we can evaluate the IFF objective $\mathfrak{F}$.

Notably, $\nu^2 \in \mathbb{R}_{\geq 0}$ depends only on $y$, so can be precomputed and stored with only $O(N)$ cost, and $\bar{y} \in \mathbb{R}^M$ depends only on $x, y$ and $z$, so can also be precomputed and stored with $O(NM)$ cost. Similarly, the matrix $K_{u\mathfrak{f}}K_{u\mathfrak{f}}^*$ can be precomputed with $O(NM^2)$ cost. For large $N$, we split the data into chunks of $10\,000$ to save memory in this precompute stage. During optimisation, each calculation of this objective is reduced to the $O(M^3)$ cost associated with the inverse and log determinant calculations, which we perform cheaply after first performing the Cholesky decomposition.

The prediction equation is

$$q(f(x_*)|x_*, x, y) = \int p(f(x_*)|x, z, u)q(u)du = \mathcal{N}(f(x_*)|K_{u*}^* K_{uu}^{-1}\mu_u, K_{**} - K_{u*}^* K_{uu}^{-1}K_{u*} + K_{u*}^* K_{uu}^{-1}\Sigma_u K_{uu}^{-1}K_{u*}) \tag{29}$$

where $*$ stands for $x_*$ in the subscripts. This is just the sparse, inter-domain version of Equation (2), and $\mu_u, \Sigma_u$ are given in Equation (7).

## B  Real-valued features

As noted in Section 4.1, applying an invertible linear transformation to the features does not change inference or learning.

To simplify the presentation, and generalise the result, we change notation slightly from the main text. Let the complex valued featured be $u'_{m_1,\ldots,m_D}$ which is the feature corresponding to the frequency $z_{m_1,\ldots,m_D}$, and let $z$ be antisymmetric in every axis (that is, $z_{-m_1,m_2,\ldots,m_D} = -z_{m_1,m_2,\ldots,m_D}$, etc) and require $m_d \neq 0$ for each $d$.

For example, if we use a regular grid, with $M^{1/D}$ an integer,

$$z_{m_1,\ldots,m_D} = \begin{bmatrix} \varepsilon/2 + m_1\varepsilon - M^{1/D}/2 \\ \vdots \\ \varepsilon/2 + m_D\varepsilon - M^{1/D}/2 \end{bmatrix}$$

which indeed satisfies this property.

Now,

$$
\begin{aligned}
e^{-i2\pi x^\top \xi} &= \cos(2\pi x^\top \xi) + i\sin(2\pi x^\top \xi) \\
&= \sum_{j=0}^{D}(-i)^j \sum_{S\subseteq\{1:D\},|S|=j} \prod_{d\in S}\sin 2\pi x_d z_d \prod_{d\notin S}\cos 2\pi x_d z_d
\end{aligned}
$$

so let $u_{m_1,m_2,\dots,m_D,S}$ for positive $m_d$ only, with $S\subseteq\{1:D\}$, be defined as

$$
u_{m_1,\dots,m_D,S} = \int_{z_D-\varepsilon/2}^{z_D+\varepsilon/2}\cdots\int_{z_1-\varepsilon/2}^{z_1+\varepsilon/2}\int f(x)\prod_{d\in S}\sin 2\pi\xi_d x_d\prod_{d\notin S}\cos 2\pi\xi_d x_d\,dxd\xi_1\dots d\xi_D
$$

so that

$$
u'_{m_1,\dots m_D} = \sum_S (-i)^{|S|}u_{|m_1|,\dots,|m_D|,S}\prod_{d\in S}\operatorname{sgn}(m_d).
$$

**Lemma B.1.** *The real representation is equivalent to the complex representation used the main text.*

*Proof.* It suffices to show that this transformation can be expressed as a matrix with linearly independent rows. The row corresponding to each $u'_{m_1,\dots,m_D}$ has only non-zero entries for $u_{|m_1|,\dots,|m_D|,S}$ for any $S$. Hence, if the absolute values of $m_1,\dots,m_D$ differ, then the rows are linearly independent. Now, suppose the absolute values are fixed and consider an arbitrary collection of indices $S'\subseteq\{1:D\}$ to have negative sign. This corresponds to a particular $u'$, hence a particular row of the matrix. Then the non-zero entries in the row corresponding to $S'$ each correspond to a choice of $S$, and the sign is flipped (relative to the case where each $m_d$ is positive) if $|S\cap S'|$ is odd.

The rows are only linearly dependent if they have all their signs flipped or all their signs not flipped. That is, if there exists $S'\neq S''$ such that $|S\cap S''|$ and $|S\cap S'|$ have the same parity for all $S\subseteq\{1:D\}$. But since they are distinct, there must be at least one $d\in\{1:D\}$ which is in $S'$ but not $S''$ (or vice versa) and so for $S=\{d\}$, the parity differs. $\square$

## C  Convergence

We follow the steps of Section 4.1, generalising to higher dimensions and filling in the details.

Recall that inference and learning with with the modified features $u_3$ and their approximation $\hat{u}_3$ is equivalent to inference and learning with the features we use in practice. In this section, for brevity, we use $\hat{u}=\hat{u}_3$, and $u=u_3$.

Following from Burt et al. (2019), the gap between the log marginal likelihood and the training objective is bounded as

$$
\mathbb{E}_y[D_{KL}(q(f)\,||\,p(f|y))] = \mathbb{E}_y[\mathcal{L}-\mathcal{F}] \leq \frac{t}{\sigma^2} \tag{30}
$$

$$
t = \operatorname{tr}(K_{\mathrm{ff}} - \underbrace{K_{uf}^* K_{uu}^{-1}K_{uf}}_{Q_{\mathrm{ff}}}) \tag{31}
$$

when $u$ are valid inducing features, and the data $y$ are generates according to the Equation (1). We defer the effect of approximating with $\hat{u}$ until later in the proof. First, we set out the technical assumptions.

Let $k=v\sigma^2\tilde{k}$ where $\tilde{k}(x,x)=1$ (that is, define $v$ as the signal to noise ratio).

**A1**  Assume that $\tilde{k}$'s spectral measure admits a density, and denote this by $\tilde{s}$ and assume that the density admits a tail bound

$$
\int_\rho^\infty\cdots\int_\rho^\infty \tilde{s}(\xi)\,d\xi_1\dots d\xi_D \leq \beta\rho^{-qD} \tag{32}
$$

for any $\rho>0$ and some $\beta,q>0$.

**A2**  The spectral density's second derivative is bounded.

**A3**  The spectral density's first derivatives are bounded as

$$\frac{\partial s(\xi)}{\partial \xi} \leq 2Ls(\xi) \implies \frac{\partial \sqrt{s(\xi)}}{\partial \xi} \leq L\sqrt{s(\xi)} \tag{33}$$

for some $L > 0$ (where the second expression follows wherever $s(\xi) > 0$). For example, this would be satisfied if the spectral density and its first derivative are both bounded everywhere, which includes widely used covariance functions.

**A4**  Finally, let the inducing frequencies $z_m = (-(M+1)/2 + m)\varepsilon$ in one dimensions, and an analogous regular grid in higher dimensions, with $M^{1/D} \in \mathbb{Z}$ even. That is, for a multi-index $m_{1:D}$,

$$[z_{m_{1:D}}]_d = (-(M^{1/D}+1)/2 + m_d)\varepsilon \quad \text{for } d \in \{1:D\}. \tag{34}$$

Notationally, we usually use a single index $m$ and, for brevity, let

$$\square_m = \prod_{d=1}^{D} [[z_m]_d - \varepsilon/2, [z_m]_d + \varepsilon/2), \quad \square = \bigcup_m \square_m \tag{35}$$

We might wish to deviate from A4 in practice – for example, to prioritise higher importance frequencies when using a finite number, or to vary $\varepsilon$ in each dimension or as a function of location. We note that the proofs which follow can be generalised to these cases, with the rate of convergence controlled by the largest width $\varepsilon$ used.

In the rest of this section, we refer to these as the standard assumptions, and we now reiterate the definitions of $u$ and $\hat{u}$.

$$u = \langle \phi_{3,m}, f \rangle \overset{\text{def}}{=} \varepsilon^{-1} \int_{\square_m} \int f(x) e^{-i2\pi\xi x} \, dx / \sqrt{s(\xi)} \, d\xi = \varepsilon^{-1} \int_{\square_m} \langle \phi_{1,\xi}, f \rangle \sqrt{s(\xi)} \, d\xi$$

$$c_m(x') = c_3(x', z_m) = \varepsilon^{-1} \int_{\square_m} c_1(x', \xi) \sqrt{s(\xi)} \, d\xi \approx \sqrt{s(z_m)} e^{-i2\pi z_m x'} = \hat{c}_m(x')$$

$$\bar{k}_{m,m'} = \bar{k}_3(z_m, z_{m'}) = \varepsilon^{-2} \int_{\square_m} \int_{\square_{m'}} \bar{k}_1(\xi, \xi') s(\xi) \, d\xi' \, d\xi = \varepsilon^{-1} \delta_{m-m'}$$

Then $\hat{u}$ is defined implicitly through the definition of $\hat{c}_m, \bar{k}_{mm}$ above.

**Lemma C.1** (Lemma 4.1). *Under assumptions A3 and A4,*

$$c_m(x) = \hat{c}_m(x)(1 + O(\varepsilon^D)) \tag{36}$$

*Proof.* We first consider the 1D case. Let $\mathcal{E}_m(\xi) = \sqrt{s(\xi)} - \sqrt{s(z_m)}$. Then by Taylor's theorem, $|\mathcal{E}_m(\xi)| \leq L|\xi - z_m|\sqrt{s(\xi')}$ for some $\xi' \in \square_m$. In particular, let $\xi' = \arg\max_{\xi \in \square_m} \sqrt{s(\xi)}$. But we also have $\sqrt{\xi'} \leq \sqrt{z_m} + L\sqrt{s(\xi')}|\xi' - z_m| \leq \sqrt{z_m} + L\sqrt{s(\xi')}\varepsilon/2$. Then for $\varepsilon < 2/L$ it follows that

$$\sqrt{s(\xi')} \leq \sqrt{s(z_m)} \frac{1}{1 - \frac{L\varepsilon}{2}} = \sqrt{s(z_m)}(1 + O(\varepsilon)) \tag{37}$$

and so

$$|\mathcal{E}_m(\xi)| \leq L|\xi - z_m|\sqrt{s(z_m)}(1 + O(\varepsilon)). \tag{38}$$

Then, considering at first the upper bound

$$c_m(x) = \varepsilon^{-1} \int_{\square_m} \sqrt{s(\xi)} e^{-i2\pi\xi x} \, d\xi \tag{39}$$

$$= \varepsilon^{-1} \int_{\square_m} (\sqrt{s(z_m)} + \mathcal{E}_m(\xi)) e^{-i2\pi z_m x} e^{-i2\pi(\xi - z_m)x} \, d\xi \tag{40}$$

$$= \sqrt{s(z_m)} e^{-i2\pi z_m x} \varepsilon^{-1} \int_{\square_m} e^{-i2\pi(\xi - z_m)x} \, d\xi + \varepsilon^{-1} \int_{\square_m} \mathcal{E}_m(\xi) e^{-i2\pi\xi x} \, d\xi \tag{41}$$

$$\leq \hat{c}_m(x)\mathrm{sinc}(2\pi\varepsilon x) + \varepsilon^{-1} \int_{\square_m} |\mathcal{E}_m(\xi) \, d\xi \tag{42}$$

$$\leq \hat{c}_m(x) \left( \mathrm{sinc}(2\pi\varepsilon x) + \frac{L\varepsilon}{2} \right) \tag{43}$$

Here $\mathrm{sinc}(\alpha) = \sin(\alpha)/\alpha$. The sinc term is of constant order. The lower bound is found by subtracting the magnitude of the error term instead of adding. The result then follows.

In higher dimensions, we follow the same argument, and the new upper bound is

$$c_m(x) \leq \hat{c}_m(x) \left( \mathrm{sinc}^D(2\pi\varepsilon x) + \frac{L\varepsilon^D}{2} \right) \tag{44}$$

from which the result follows. $\qquad\square$

*Remark* C.2. The error bound in Equation (44) generally tightens as $\varepsilon$ falls, but loosen as $x$ increases, with the approximation value vanishing when $\varepsilon = 1/x$.

**Theorem C.3** (Theorem 4.2 of the main text). *Under the assumptions A1-A4, for any $\Delta, \delta > 0$, there exists $M_0, \alpha_0 > 0$ for all $M > M_0$,*

$$\mathbb{P}\left[ \frac{\mathcal{L} - \mathfrak{F}}{N} \geq \frac{\Delta}{N} \right] \leq \delta$$

*and with*

$$M \leq \left( \frac{\alpha}{\Delta\delta} N \right)^{\frac{q+3}{2q}} .$$

*Proof.* Let $\hat{t} = \mathrm{tr}(K_{\mathrm{ff}} - \hat{K}_{u\mathrm{f}}^* K_{uu}^{-1} \hat{K}_{u\mathrm{f}}) = \mathrm{tr}(K_{\mathrm{ff}} - \hat{Q}_{\mathrm{ff}})$. Then we can show that $\hat{t}/N\sigma^2 \in O(M^{-2q/(q+3)})$ as follows.

$$\frac{\hat{t}}{N\sigma^2} = \frac{1}{N\sigma^2} \sum_n \left( \underbrace{k(x_n, x_n)}_{v\sigma^2} - \varepsilon^D \sum_m \hat{c}_m(x_n)\hat{c}_m^*(x_n) \right) \tag{45}$$

$$= v \left( 1 - \varepsilon^D \sum_m \tilde{s}(z_m) \right) \tag{46}$$

$$= v \left( 1 - \int_\square \tilde{s}(\xi) \, d\xi \right) + vE_1 \tag{47}$$

$$= 2v \int_{\mathbb{R}\backslash\square} \tilde{s}(\xi) \, d\xi + vE_1 \tag{48}$$

where $E_1 = \int_\square \tilde{s}(\xi) \, d\xi - \varepsilon \sum_m \tilde{s}(z_m) + O(M\varepsilon^{3D})$. The integral term in the last line is in $O((M^{1/D}\varepsilon)^{-qD})$ by assumption A1, and $E_1 \in O(M\varepsilon^{3D})$ from standard bounds on the error of the midpoint approximation (since the second derivative of the integrand is bounded by assumption A2).

We must have that $\varepsilon \to 0$ as $M \to \infty$ to make the midpoint approximation asymptotically exact, yet we must have $M\varepsilon \to \infty$ to ensure the features cover all frequencies. We optimise the trade-off between these two.

In particular, let $\varepsilon = \varepsilon_0 M^{-p/D}$ for some $\varepsilon_0 > 0$ and $p \in (1/3, 1)$. Then,

$$\frac{\hat{t}}{N\sigma^2} \in O(M^{-q(1-p)} + M^{-(3p-1)}). \tag{49}$$

The overall rate is asymptotically dominated by the worse of these two rates, so we optimise $p$ as

$$p = \arg\max_{p'} \min\{q(1-p), 3p-1\} = \frac{q+1}{q+3} \tag{50}$$

which is the $p$ which sets both rates equal at

$$\frac{2q}{q+3}.$$

Altogether we have

$$\frac{\hat{t}}{N\sigma^2} \in O(M^{\frac{-2q}{q+3}}). \tag{51}$$

For bounding $\mathbb{E}[(\mathcal{L} - \mathcal{F})/N]$ in terms of $t$, we could immediately apply the result Equation (30). But for $\mathbb{E}[(\mathcal{L} - \mathfrak{F})/N]$ we cannot, since $\hat{u}$ are not exact features. But following the proof of Lemma 2 of Burt et al. (2019), we have

$$\mathbb{E}_y \left[ \frac{\mathcal{L} - \mathfrak{F}}{N\sigma^2} \right] \leq \frac{\hat{t}}{N\sigma^2} + O(M\varepsilon^{3D}) \tag{52}$$

provided $\hat{Q}_{\mathrm{ff}} \geq 0$ and $(\log |\hat{Q}_{\mathrm{ff}} + \sigma^2 I| - \log |K_{\mathrm{ff}} + \sigma^2 I|)/N\sigma^2 < O(M\varepsilon^{3D})$.

For the first condition, apply Bochner's theorem. Consider the covariance function

$$q(x, x') = \sum_m \hat{c}_m(x)\hat{c}_{m'}^*/\hat{k}_{mm'} \tag{53}$$

which is used to form the elements of $\hat{Q}_{\mathrm{ff}}$. Its Fourier transform is

$$[\mathcal{F}q](\xi) = \varepsilon \sum_m s(z_m)\delta(\xi - z_m) \tag{54}$$

which is indeed a positive measure, so $q$ is a positive definite covariance function, so $\hat{Q}_{\mathrm{ff}} \geq 0$.

For the log determininant term, we have from Lemma 4.1 (which holds due to A3, A4) that the relative error of $\hat{Q}_{\mathrm{ff}}$ from $Q_{\mathrm{ff}}$ is symmetric and in $O(M\varepsilon^{3D})$, hence the error in the log determinant is in $O(NM\varepsilon^{3D})$; the scaled identity shift does not change the order of this relative error. Then,

$$(\log |\hat{Q}_{\mathrm{ff}} + \sigma^2 I| - \log |K_{\mathrm{ff}} + \sigma^2 I|)/N\sigma^2 \leq (\log |Q_{\mathrm{ff}} + \sigma^2 I| - \log |K_{\mathrm{ff}} + \sigma^2 I|)/N\sigma^2 + O(M\varepsilon^{3D}) \tag{55}$$
$$\in O(M\varepsilon^{3D}) \tag{56}$$

where the last step follows from $(\log |Q_{\mathrm{ff}} + \sigma^2 I| - \log |K_{\mathrm{ff}} + \sigma^2 I|) \leq 0$ (see proof of Lemma 2 of Burt et al. (2019)). Thus, we have

$$\mathbb{E}_y \left[ \frac{\mathcal{L} - \mathfrak{F}}{N} \right] \in O(M^{-\frac{2q}{q+3}}) \tag{57}$$

Then the first part of the results follow by a straightforward application of Markov's inequality. For the second part, we replace the big $O$ notation with an explicit constant $\alpha_0 > 0$.

$$\mathbb{P}\left[ \frac{\mathcal{L} - \mathfrak{F}}{N} \leq \frac{\Delta}{N} \right] = \delta \leq \frac{\mathbb{E}[(\mathcal{L} - \mathfrak{F})/N]}{\Delta/N} \tag{58}$$

$$\leq N\frac{\alpha}{\Delta}M^{-\frac{2q}{q+3}} \tag{59}$$

$$\implies M \leq \left( \frac{\alpha}{\delta\Delta}N \right)^{\frac{q+3}{2q}} \tag{60}$$

$$\square$$

This result demonstrates that the objective we use for hyperparameter optimisation, even with the numerical approximation, converges at a reasonable rate to the log marginal likelihood, at least if the spectral density has sufficiently light tails. This means it is a good surrogate for learning when we can set $M$ large enough.

With the proper inducing features $u$, we can be reassured that the posterior predictives will not be too bad, since the whole process KL from the approximating to exact posterior is bounded as $\mathcal{L} - \mathcal{F}$ Matthews et al. (2016). With $\hat{u}$, we require some additional reassurance, which the is the subject of the next result.

**Theorem C.4** (Theorem 4.4 of the main text)**.** *For the optimised $\mu_u, \Sigma_u$ (to maximise $\mathfrak{F}$), let the posterior predictive at any test point $x_*$ using the exact features $u$ have mean and variance $\mu, \Sigma$, and with the approximate features $\hat{u}$ have mean and variance $\hat{\mu}, \hat{\Sigma}$. Then, under assumptions A3 and A4,*

$$|\mu - \hat{\mu}| \in O(M\varepsilon^{2D}) \tag{61}$$

$$|\Sigma - \hat{\Sigma}| \in O(M^2\varepsilon^{3D}) \tag{62}$$

*and in particular, allowing $\varepsilon \sim M^{-\frac{q+1}{D(q+3)}}$ as in the proof of Theorem C.3, we have*

$$|\mu - \hat{\mu}| \in O\left(M^{-\frac{q-1}{q+3}}\right) \tag{63}$$

$$|\Sigma - \hat{\Sigma}| \in O\left(M^{-\frac{q-3}{q+3}}\right) \tag{64}$$

This result shows that the predictive marginals using the approximation converge at a reasonable rate to those without the approximation, for any fixed $\mu_u, \Sigma_u$ as long as the spectral density is not too heavy tailed ($q > 3$). We note that we do not comment on the rate at which the optimal variational means and covariances converge to each other, nor the consequent rate of convergence for the posterior predictives according to each objective's optimal variational distribution.

*Proof.* The predictive distributions are

$$q(f(x_*)) = \mathcal{N}(f(x_*)| \underbrace{K_{u*}^* K_{uu}^{-1}\mu_u}_{\mu}, \underbrace{k(x_*, x_*) - K_{u*}^* K_{uu}^{-1}K_{u*} + K_{u*}^* K_{uu}^{-1}\Sigma_u K_{uu}^{-1}K_{u*}}_{\Sigma} \tag{65}$$

$$\hat{q}(f(x_*)) = \mathcal{N}(f(x_*)| \underbrace{\hat{K}_{u*}^* K_{uu}^{-1}\mu_u}_{\hat{\mu}}, \underbrace{k(x_*, x_*) - \hat{K}_{u*}^* K_{uu}^{-1}\hat{K}_{u*} + \hat{K}_{u*}^* K_{uu}^{-1}\Sigma_u K_{uu}^{-1}\hat{K}_{u*}}_{\hat{\Sigma}}. \tag{66}$$

Recall that $K_{uu}^{-1} = \varepsilon^D I$, and that $|[K_{u*} - \hat{K}_{u*}]_m| = |c_m(x_*) - c_m(x_*)| \in O(\varepsilon^D)$ by Lemma 4.1. The result for the means follows straightforwardly.

$$|\mu - \hat{\mu}| = |(K_{u*}^* - \hat{K}_{u*}^*)K_{uu}^{-1}\mu_u| \in O(M\varepsilon^{2D}) \tag{67}$$

which follows since $K_{uu}^{-1}\mu_u$ is an $M$ dimensional vector with each element in $O(\varepsilon^D)$. For the covariance, we use the triangle inequality.

$$|\Sigma - \hat{\Sigma}| \leq |K_{u*}^* K_{uu}^{-1}K_{u*} - \hat{K}_{u*}^* K_{uu}^{-1}\hat{K}_{u*}| \tag{68}$$

$$+ |K_{u*}^* K_{uu}^{-1}\Sigma_u K_{uu}^{-1}K_{u*} - \hat{K}_{u*}^* K_{uu}^{-1}\Sigma_u K_{uu}^{-1}\hat{K}_{u*}| \tag{69}$$

Now, each of the terms on the right hand side is a one dimensional marginal variance, so we rewrite them as follows.

$$|K_{u*}^* K_{uu}^{-1}K_{u*} - \hat{K}_{u*}^* K_{uu}^{-1}\hat{K}_{u*}| = \varepsilon^D \sum_m (|c_m(x_*)|^2 - |\hat{c}_m(x^*)^2|) \tag{70}$$

$$\in O(M\varepsilon^{2D}) \tag{71}$$

$$|K_{u*}^* K_{uu}^{-1}\Sigma_u K_{uu}^{-1}K_{u*} - \hat{K}_{u*}^* K_{uu}^{-1}\Sigma_u K_{uu}^{-1}\hat{K}_{u*}| = \varepsilon^{2D} \sum_{m,m'} [\Sigma_u]_{m,m'} (c_m(x_*)c_{m'}^*(x_*) - \hat{c}_m(x_*)\hat{c}_{m'}^*(x_*)) \tag{72}$$

$$\in O(M^2\varepsilon^{3D}) \tag{73}$$

The first term is $\varepsilon \sum_m(|c_m(x_*)|^2 - |\hat{c}_m(x^*)^2|) \in O(M\varepsilon^{3D})$ following from the proof of Theorem C.3. The other two terms are of the same order $O(M\varepsilon^{2D})$ using Lemma 4.1. $\qquad \square$

# D Experimental Details

We include the code for the experiments and figures which can be referred to for full details.

For Figure 2, we used $N = 1\,000$ data points, whose input locations were sampled from a zero mean Gaussian distribution with standard deviation $W_x/6$. We randomly sampled a signal standard deviation $\sigma_f \approx 0.5422$ and a signal to noise ratio (SNR) $v \approx 1.255$, and used a unit lengthscale ($\lambda = 1$). To verify that the pattern is broadly consistent for different parameter choices, we also reproduced the figure with lengthscale $\lambda = 0.5$, and, with both lengthscales, we used both half and four times the SNR. In each case, we set $W_x = 6\lambda^{-1}\sqrt{N/2}$. FInally, we consider the case with uniformly sampled data. The results are plotted in Figures 7 to 12.

For the synthetic experiment, we generated $N = 10\,000$ data points in 1 and 2 dimensions by samping from a GP with a Gaussian or Matérn-5/2 covariance function, with unit (or identity) lengthscale, unit variance, and set the SNR to 0.774 (arbitrarily chosen, poor signal to noise ratio for a challenging dataset). In 1D we sample the training inputs uniformly on a width $6\sqrt{N/2}$ centred interval. In 2D we do the same in each dimension, but with width 5. We then fit each model plotted, training using LBFGS and using the same initialisation in each case, other than necessary restrictions on the choice of covariance functions as described in the main text. The initial values were lengthscales of 0.2, and unit signal and noise variances. We did multiple random trials, and plot the 2 standard deviation error bars; this is mainly for uncertainty in timing, but for inducing points we also have uncertainty due to randomness in the inducing point (re)initialisation method. Inducing points with inducing inputs optimised takes far longer than the other methods, so was excluded.

For the real-world experiments, we used a similar setup as the synthetic experiments in terms of initialisations. Guided by the synthetic results, we use the full rectangular grid of frequencies for VFF, but use a spherical mask for IFF. We set $\varepsilon$ as described in the main text. Comparably, for B-splines and VFF, we set the interval $[a, b]$ as 0.1 wider than the data (that is, $a_d = \min_{n,d} x_{n,d} - 0.1$, $b_d = \max_{n,d} x_{n,d} + 0.1$); we use fourth order B-splines. The experiments were generally run on CPU to avoid memory-related distortion of the results, with the exception of SKI, which was run on GPU since it depends on GPU execution for faster MVMs.

We implemented IFF, VFF and spherical harmonics using `gpflow` (Matthews et al., 2017), which we also used for inducing points, reusing some of the code of Burt et al. (2020b) for the $k$-DPP (re)initialisation method. The spherical harmonics implementation depends on the backend-agnostic implementation of the basis functions[3], though as noted in the main text, we were unable to produce comparable results. We used the `gpytorch` implementation of SKI (Gardner et al., 2018b). We use the publicly available Tensorflow 2 implementation for B-splines[4].

## D.1 Real-world dataset information

In all cases, we normalise both the inputs and targets to unit mean and standard deviation in each dimension. However, we report the test metrics (RMSE and NLPD) averaged over test points but on the unnormalised scale. The number of training and test points, and the standard deviation of the outputs, for each dataset is reported in Table 1

The precipitation dataset is a regularly gridded (in latitude and longitude) modelled precipitation normals in mm in the contiguous United States for 1 January 2021 (publicly available with further documentation at `https://water.weather.gov/precip/download.php`; note the data at the source is in inches). We downsample the data by 4 (by 2 in each dimension). The data is highly nonstationary, and so a challenging target for GP regression with typically used, usually stationary, covariance functions. In particular, the lengthscales are fairly large across the plains and in the southeast in general, but quite small near the Pacific coast, especially in the Northwest. For a stationary model, the high frequency content in that region leads to a globally low lengthscale.

---

[3] `https://github.com/vdutor/SphericalHarmonics`
[4] `https://github.com/HJakeCunningham/ASVGP`

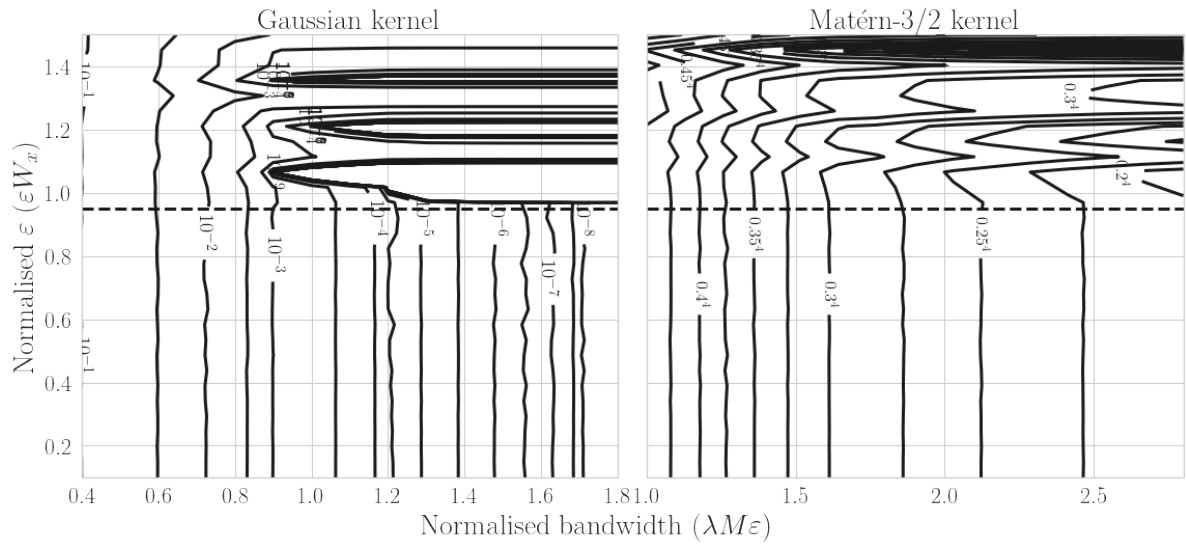

Figure 7: As Figure 2, but with half the SNR.

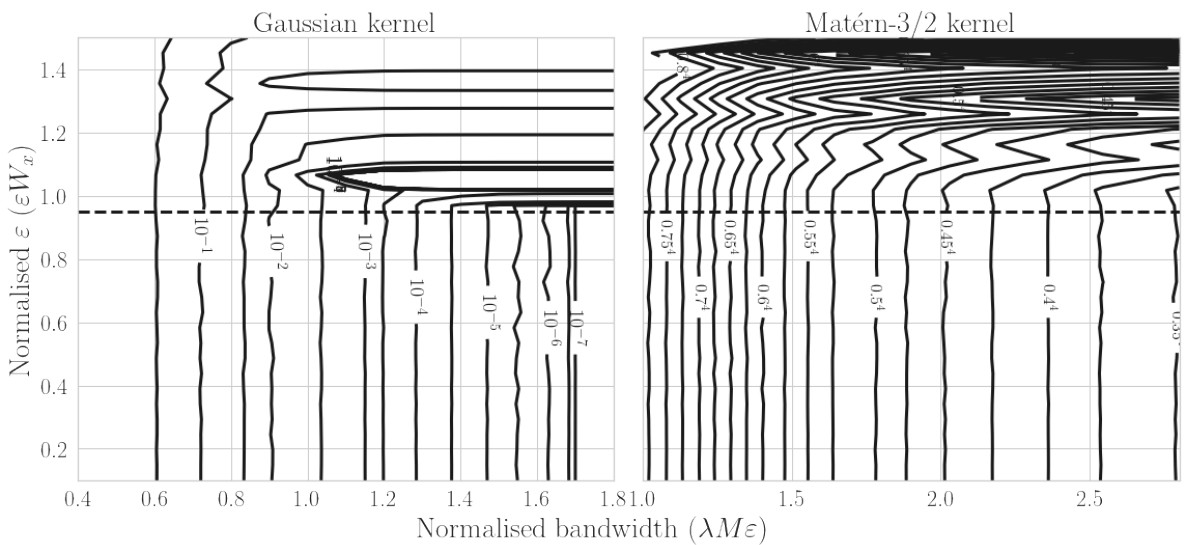

Figure 8: As Figure 2, but with four times the SNR.

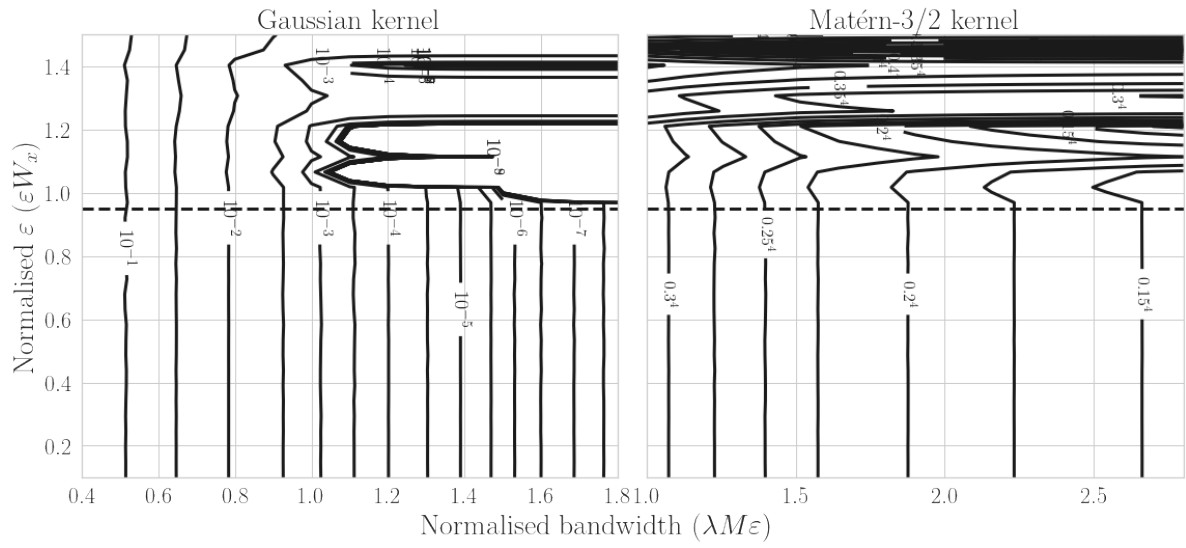

Figure 9: As Figure 2, but with half the lengthscale (twice the bandwidth).

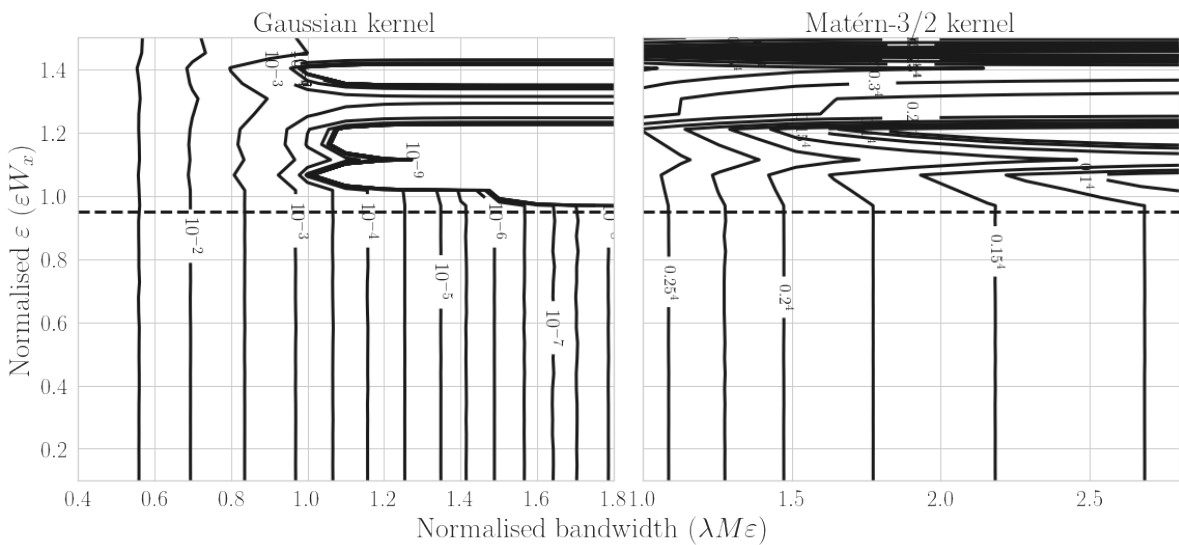

Figure 10: As Figure 2, but with half the lengthscale (twice the bandwidth) and half the SNR.

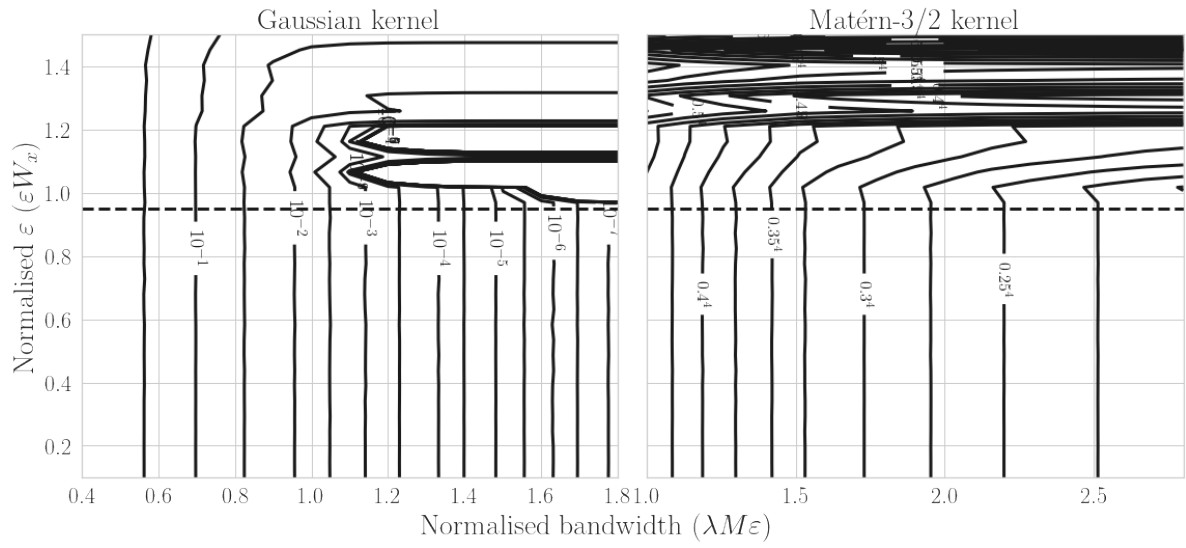

Figure 11: As Figure 2, but with half the lengthscale (twice the bandwidth) and four times the SNR.

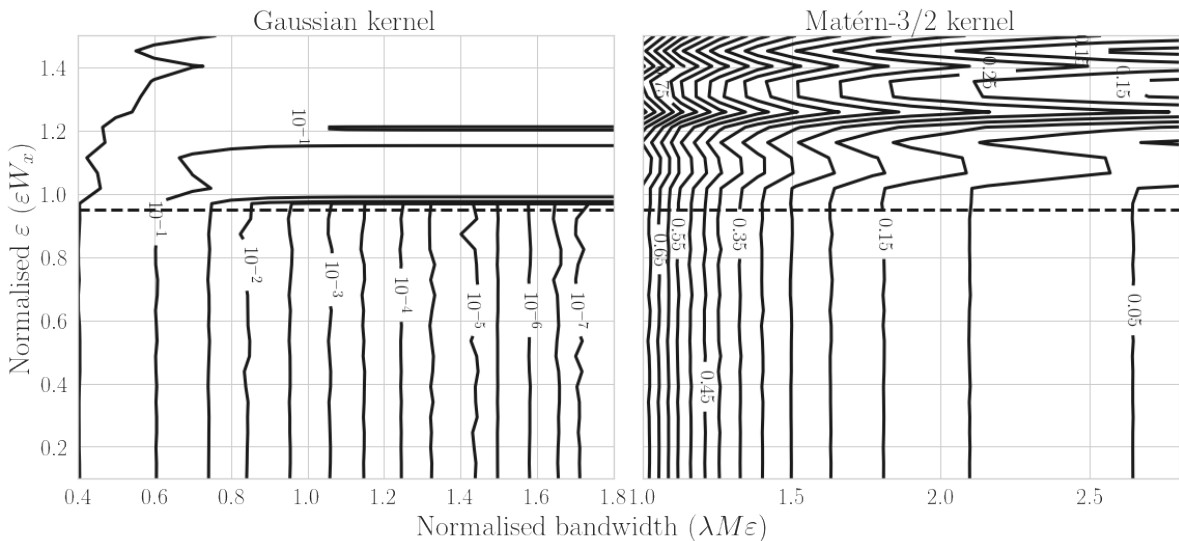

Figure 12: As Figure 2, but with data sampled uniformly on $[-W_x/2, W_x/2]$, and with higher SNR ($\approx 2.067$).

Table 1: Summary of real world datasets.

| Dataset | $N$ | Test points | Output standard deviation |
|---|---|---|---|
| Temperature | 12 947 | 3236 | 2.766 |
| Precipitation | 23 144 | 5785 | 58.855 |
| House price | 106 875 | 26 718 | 0.642 |

The temperature dataset is the change in mean land surface temperature (°C). over the year ending February 2021 relative to the base year ending February 1961 (publicly available from `https://data.giss.nasa.gov/gistemp/maps`). It is also regularly gridded, over more of the globe.

The house price dataset is a snapshot of house prices in England and Wales, which is not regularly gridded. We use a random 20% of the full dataset, and target the log price to compress the dynamic range. It is based on the publicly available UK house price index (`https://landregistry.data.gov.uk/app/ukhpi`), and we enclose the exact dataset we use.

