# OpenReview forum: "Integrated Variational Fourier Features for Fast Spatial Modelling with Gaussian Processes"
_TMLR — Accepted by TMLR_

### Review · Reviewer_ZPJr · 2023-10-06

**Summary Of Contributions:**

The paper introduces integrated Fourier features for Gaussian processes (GPs) to enhance their scalability for larger datasets. The method extends performance benefits to a broad class of stationary covariance functions. Overall speaking, this paper proposed a speedup method for Gaussian process applied on large dataset. Experiments show that the proposed IFF method can be applied on a various source of dataset and show great scalability, reducing the  computational costs especially when M << N.

**Audience:**

Yes

**Broader Impact Concerns:**

There doesn't seem to be any direct ethical concerns related to the paper's content.

**Claims And Evidence:**

Yes

**Requested Changes:**

I think there is nothing much to change. Maybe it will be good to have a table to compare the computational costs and other properties of current methods for solving Gaussian process problem.

**Strengths And Weaknesses:**

Strength:
1. The proposed features are motivated by a convergence analysis and empirical exploration.
2. The paper demonstrates practical speedup in synthetic and real-world spatial regression tasks.

Weakness:
1. The paper's experiments involve synthetic datasets with arbitrarily chosen parameters, such as a signal-to-noise ratio (SNR) of 0.774, which may not be representative of real-world scenarios.
2. The paper mentions inducing points with inducing inputs optimized taking longer than other methods, but it does not delve deeper into the implications of this observation.

---

### Review · Reviewer_y6ZF · 2023-12-01

**Summary Of Contributions:**

The work proposes integrated Fourier features to perform fast-and-scalable GP regression in O(M^3) with stationary kernels and of regular spectral density (i.e. non-periodic kernels). The proposed ideas build on top of inter-domain GPs and recent methods in the context of frequency-based representations of kernels to obtain scalable sparse approximations for GPs. While following this line of thinking, the propositions show a good performance on the experimental results on both synthetic and real-world datasets.

**Audience:**

Yes

**Claims And Evidence:**

Yes

**Requested Changes:**

Some changes I would like to see updated to vote for a strong acceptance of the paper and likely for improving its clarity and impact in the future.

**Point 1.** Page (1) --- "the scaling with N is problematic ...." I see the point of saying this and reasoning that scaling with N and using stochastic optimization makes the learning process slower in general. However, if the paper needs this sort of thinking for avoiding the use/introduction of stochastic gradient descent methods, I think more attention should be put on the computational advantage of the method, like really showing the time-difference between using Fourier features with sparse approximations and other stochastic gradient-based methods. I basically say this bc many readers are now familiar and native in the use of SGD, even with GPs, and stating that the proposed ideas are much more faster, even showing clear empirical results would convince many of them to believe this is a good direction. Indeed, this last sort of things would be better than the last paragraph.

**Point 2.** Fixing writing typos. I caught quite a few writing typos like "pesudo" instead of pseudo or "for for" just in the first pages. A deep look into the text and some rewording/rewriting of such typos would help.

**Point 3.** The introduction is great in general, but when one jumps in the background things get a bit messy. So it took me quite a few reads to remember/realize the key point of everything, which is that Kuu is diagonal because the Fourier features are independent. Of course, this in the end is what makes the computation to be reduced to O(M^3) bc the trace can be reduced down to the equation written in Appendix A. This point is kind of hidden in the text, as one realizes it when looking into Appendix A. Just notice that this Kuu is usually the GP prior on standard inducing points, so any practitioner might accidentally think that Kuu is still fully correlated and then have issues to understand what is going on. So I think is important to remark and remind all around the paper that this Kuu is diagonal and features independent in all cases.

**Point 4.** Could the notation be matched with D. Burt's papers or VFF? I think is not a lot of work, (i.e. using bold letters for inducing features and matrices for instance and maybe using more standard acronyms for frequency domains), and it would save a lot of thinking-time for the reader.

**Point 5.** Some lingering questions in my mind that maybe are obvious but could be perhaps clarified. The issues with the Dirac delta: a) unsuitable for constructing the conditional prior (this needs more explanations and at least to be smoother for the reader), and b) difficulties to integrated out the Dirac delta in Eq. (11) --- is this bc it cannot be done, not numerical methods available, intractability, difficult functional evaluation, etc... the reasons are not clear to me here and i think is super important.

**Point 6.** Spectral density, c and k. These three elements are the ones presented and indexed at every step of the paper. Indeed, the different methods have different indices 1,2,3.. However, this is a bit difficult to follow and not very clear. Perhaps, it would help to cleary state the role of every one of them, and also use other way to say that they are different approximations or methods.

**Point 7.** I see the utility of averaging the Fourier features over disjoint intervals of fixed width. But as said before, this is very related to the problems on integrating out the Dirac delta and so on. Could it be explained a bit more why the averaging wants disjoint intervals, if this ones are exactly near each other, the size of \epsilon chosen, etc...?

**Point 8.** The transformation of the features by the invertible linear transformation T seems useful, but I do not understand what is going on and how it helps to have only one approximation... This part is super difficult to follow for me, and also do not understand why the normalization by spectral density turns into the normalization by square root. I guess bc T is used, but not clear how to me.

**Point 9.** Why only for D<4? I see that it performs poorly for larger D, but where is the bottleneck or issue? I would also add a bit more of analysis/detailed context in the results, and connecting things a bit better with the Appendix. For instance, explaining a bit the nature of the real-world datasets and the characteristics of the data to easily understand the impact of the performance.

**Strengths And Weaknesses:**

**Strengths.**

Overall, I think the paper introduces novel ideas and relevant advances for obtaining scalable approximations for GP regression. I particularly liked how the authors took and presented ideas from previous works in the context that they were interested in before jumping into their main technical propositions. Despite the derivations with Fourier features and inducing methods, I did not find any theoretical/technical typo at first sight, so the proposed equations are likely correct.

The empirical results look really good and seem to outperform previous approaches (Burt's, VFFs and standard inducing points) in precision/time, which I think is very important to remark.

I also liked that the authors clearly stated the limitations of the method and the gaps that they left for further development or just bc they were out of the scope of the current manuscript. In that direction, I think the work is extremely transparent with the ideas and the justification of the decisions taken, which is important for its future impact and understanding of readers.

**Weaknesses.**

Having stated the main strengths, and once again remarking that I think is a paper of high-quality ideas --- I think is worth saying that the manuscript should be updated to make such ideas and contributions "shine" in the text. In that regard, the paper loses clarity and some details are omitted without any reason at some points of the story. It is very clear while reading that after developing the project, some ideas are extremely clear for the writer, and then they are partially omitted or not clearly explained. These sorts of issues are of course possible to be fixed in this reviewing round, so I encourage the authors to improve the manuscript to make the paper stronger and of better clarity for its future impact in the community. For helping in this process, I will add some notes and points that I think should be improved in the next section.

---

> ### Author Response · Authors · 2024-01-26
> **Response to review**
>
> Thanks for the helpful comments. We will post a revision incorporating changes in the coming week, and will post a summary of changes with it. For now, we respond to some of the specific points raised.
>
> On point 3: diagonal $K_{uu}$ is useful for the proofs, but not for the precomputation which gives the $O(M^3$) cost. For this we only need that $K_{uf}K_{uf}^*$ (or some linear reparameterisation thereof) has no dependence on the hyperparameters. For example, in VFF (Hensman et al, 2017) or with B-spline features (Cunningham et al 2023), $K_{uu}$ is not diagonal, but one can still precompute.
>
> On point 4: What is the reviewer referring to when they write "more standard acronyms for frequency domains"?
>
> On point 5: The issue here is the one stated in the paragraph immediately following equation 11. $K_{uu}$ has unbounded diagonal, so $K_{uu}^{-1}$ vanishes -- putting this into the formulae for $p(f|u)$ shows that the conditional prior reverts to the prior.
>
> On point 7. If the intervals are not disjoint, then for any pair of features which involve integrating over intersecting intervals will be correlated, so the corresponding off-diagonal element in $K_{uu}$ would be non-zero, which would be inconvenient. The remaining points are discussed in section 4.2
>
> On point 8: in the proofs, we have to make claims about the quantity tr$K_{uf}^*K_{uu}^{-1}K_{uf} = ∑\_n∑\_m |c_m(x_n)|^2/\bar{k}_{mm}$ or its analogue under numerical approximation (using $\hat{c}$ etc). Having approximations in both the numerator and the denominator of the summand would make the calculations much more cumbersome; be linearly reparameterising we create a sequence of inducing features which have the same approximation properties (including the same convergence rate) where the we only have to worry about the impact of the numerical approximation in the numerator which significantly simplifies the calculations in the proofs.
>
> On point 9: You need the number of features to go up exponentially in dimension, and the cost is cubic in the number of features. If you double the dimensionality, you shift the performance curves substantially to the right. With something like inducing points, you really just need to select a subset of the data points, whereas with Fourier based features you need to cover a large part of the frequency domain. Depending on the specific characteristics of the dataset, you might be able to push into higher dimensions than 1-4.

---

### Review · Reviewer_bFVe · 2024-01-20

**Summary Of Contributions:**

Existing methods are either limited to kernels with specific structure (e.g., isotropic or 1D Matern kernels) or have $O(NM^2)$ computational cost, where $N$ is the sample size and $M$ is the number of features used. This work proposes a method that takes $O(M^3)$ (with $M \ll N$) per optimization step (though it was hard for me to understand the exact assumptions under which this occurs; see below).

In experiments on synthetic data ($N=10,000$), IFF consistently provides a speedup roughly of $2^5$ over standard sparse GPs (inducing points), in both 1D and 2D. In 1D, performance is similar to VFF in 1D, but significantly better in 2D. In experiments on three somewhat larger ($N=12,947$, $N=23,144$, and $N=106,875$) real datasets, IFF fairly consistently provides among the best trade-off between compute time and test performance, typically beating VFF and B-splines by a small amount.

**Audience:**

Yes

**Broader Impact Concerns:**

I have no concerns regarding ethical implications of this work.

**Claims And Evidence:**

No

**Requested Changes:**

**Major**

1. The assumptions (e.g., disjoint intervals, tail bound, first and second derivative conditions) made in Section 4 are scattered throughout the presentation, making it difficult to understand how the assumptions relate to the results and their proofs. I suggest labeling each assumption when it is first stated (e.g., A1., A2., etc.) and then explicitly referencing each assumption in the result where it is needed (e.g., instead of "Under the standard assumptions", say "Under assumptions A1 and A2").

2. As far as I can tell, the proofs of the theoretical results (Appendix C) assume the inducing features are a regular grid (Eq. (29)), but this is not clear in the main paper Section 4. In fact, the disjoint interval assumption in the first paragraph of Section 4 seems to suggest that a regular grid is not necessarily assumed. I am quite confused by this.

3. At the beginning of Section 4, it is assumed for simplicity that $D=1$, and the assumptions are stated for this case, but then the results are stated for general $D>=1$. Thus, the reader cannot understand the stated results without referring to the more general assumptions in the Appendix. I suggest sticking to one version ($D=1$ or general $D$) throughout this section.

4. Although the tail bound and second derivative assumptions on the normalized spectral density are fairly standard, I don't think the first derivative assumption (Eq. (17)) is as standard. I therefore suggest (a) adding a sentence discussing this assumption (e.g., whether is is satisfied by commonly used kernels) and (b) explicitly stating the multidimensional generalization of this assumption.

5. I couldn't quite follow how Eq. (33) follows from Eq. (32). Is there a missing "$1 +$" or something in Eq. (33)?

6. The bound on $|\Sigma - \hat{\Sigma}|$ in Theorem C.4 needs more details:
 - Why are the elements of $K_{uu}^{-1} \Sigma_u K_{uu}^{-1}$ of order O(\epsilon^D)?
 - Some more details about how Theorems C.3 and 4.1 imply the reported convergence rates of the kernel matrices $\hat K_{u*}$ and $\hat K_{u*}^*$?
 - "The other two terms are of the same order $O(M \epsilon^{2D})$": Why doesn't this $O(M \epsilon^{2D})$ doesn't dominate the overall convergence rate of $O(M \epsilon^{3D})$ in Theorem C.4? Or am I misunderstanding what "The other two terms" refers to?

7. At some points, the discussion is too vague to follow. Some examples:
 - After Eq. (15), "The first equality follows for any variationally correct features (Matthews et al., 2016).": What does "variationally correct" mean?
 - Before Eq. (14), "we use the result of Burt et al. (2019) to bound": Which result (they have several)?
As far as I could tell, these points aren't clarified in the Appendix.

**Minor**

8. Page 2, First Sentence, "iterative learning of the variational distribution which is otherwise available in closed form, which results in slower learning overall": I didn't quite understand why an iterative approach is necessarily slower than a closed form solution. For example, I think it is frequently the case that using SGD to optimize the ridge regression objective is often faster than calculating it in closed form.

9. Page 2, Last Sentence, "the hyperparameters", and elsewhere: The paper frequently refers to "the hyperparameters", but it's not clear to me what this refers to. Are there hyperparameters other than $\sigma$? Relatedly, when $\sigma$ is introduced in Eq. (1), the paper should explicitly point out if this is a hyperparameter.

10. Page 3, Sentence 1, "the quadratic form $y^⊤Ay$, both of which incur $O(N^3)$ computational cost": To clarify, the $O(N^3)$ cost is to compute $A$, right? The quadratic form itself should only take $O(N^2)$ times, right?

11. Page 9, Theorem 4.4: I suggest more explicitly writing $\mu(x_*)$, $\hat{\mu}(x_*)$, $\Sigma(x_*)$, $\hat{\Sigma}(x_*)$, to make it clearer that the quantities being bounded are scalars.

12. Figure 1: This should probably appear one page earlier, closer to where it is first referenced (near the top of page 5).

**Typos**

This paper has *many* minor typos, and needs extensive proofreading. A few examples are listed below:

13. Page 2, Paragraph 2, Last Sentence, "Existing work... Cunningham et al., 2023)": This sentence seem to parse grammatically, and I couldn't understand what it was trying to say. Is there an error here?

14. Page 2, Paragraph 3, "for for"

15. Section 3, "to approximate the solution to the linear solve"

16. Page 5, Paragraph 3, "$dxdx$"

17. Page 9, Remark 4.3, "this is due to the $O(M \epsilon^3)$ which arise due to...": It's not clear what quantity is in $O(M \epsilon^3)$. I guess this should be the error term $E_1 \in O(M \epsilon^3)$?

18. Page 11, Last Paragraph, "For out synthetic experiments"

19. Page 14, Paragraph 1, "$N = 10/, 000$"

20. Page 22, Paragraph 3, "$M^{1/D} \in $" and "That isve"

21. Page 22, Eq. (30): I think the definition of $\square_m$ is intended to be a Cartesian product, but I don't think I have ever seen $\oplus$ used for the Cartesian product. Should this be, e.g., $\prod$ or $\otimes$?

22. Page 22, defs. of $u$, $c_m$, and $\bar{k}_{m,m'}$: In the multidimensional case discussed here, I think these integrals should be over $\square_m$ rather than the 1D intervals $[z_m-\epsilon/2, z_m+\epsilon/2]$.

23. Page 22, Lemma C.1 (Theorem 4.1): I think "Theorem 4.1" should be "Lemma 4.1", consistent with Page 8.

24. Page 24, after Eq. (44), "asumptotically"

25. Page 24, after Eq. (46), "the result ??" (missing reference)

26. Page 25, "if the spectral density is sufficiently heavy tailed": I think this should be "light tailed" or "not too heavy tailed"?

27. Page 25, Theorem C.4: Should "$O(M^{−2qq+3})$" be "$O(M^{−\frac{2q}{q+3}})$"?

28. Page 25, after Eq. (63), "The first term is...": There is a missing absolute value ("$|$") sign in this expression.

**Strengths And Weaknesses:**

I think the overall goal of this paper (achieving $O(M^3)$ runtime for general kernels) seems useful and the proposed approach makes sense at a high level. However, I found the theoretical portions of this paper too unclear to verify the main results. While the topic of the paper is far from my expertise and I wasn't previously familiar with the methods (e.g., SGPR) this work is building upon, I think a substantial amount of the difficulty is due to the writing of the paper itself. Below, I ask a few questions and suggest several ways to improve the presentation.

---

> ### Author Response · Authors · 2024-01-26
> **Response to review**
>
> Thanks for the careful and detailed review and for the corrections. We will post a revision incorporating changes in the coming week, and will post a summary of changes with it. Presently, we respond to some of the specific points raised.
>
> (2) We assume a regular grid in the proofs for convenience. Some irregularities should work in practice -- one could adapt the convergence proof to several cases without too much difficulty (like different $ε$ in each dimension, and varying $ε$ as a function of $m$) as long as the intervals are kept disjoint, and that in the limit the union of all the intervals approaches $ℝ^D$. The only one of these we use in practice is to vary $ε$ by dimension. It is plausible that a future user might want to vary $ε$ locally, so that the approximation widths are narrower where the spectral density varies more quickly, or something along these lines.
>
> (3) We agree that the switch between 1D and multi D can be confusing. We think that the initial exposition is clearer in 1D, but we want to make use of the higher dimensional version, be clear that the cost goes up exponentially in dimension etc. We will move the sketch of the multidimensional generalisation, including the multidimensional assumptions into a new subsection of section 4.
>
> (4) and (5) Thanks for the comments on this part of the proof. We will amend the situation as follows. Firstly, we will move to a relative bound on the derivative: $|ds/dξ| ≤ 2L s(ξ)$. This should be satisfied for widely used stationary covariance functions, for example if the spectral density is and its first derivative are bounded everywhere. This will also slightly amend the proof of C.1 queried in (5); the revision will clarify the steps.
>
> (6) Thanks for the attention on this proof, which contains some issues. The outline of the correction is as follows (using $c'$ instead of \hat{c}). We are treating $μ_u, Σ_u$ as fixed. Also, $K_{uu}^{-1} = ε^D I$. For $K_{uu}^{-1} μ_u$, each element is scaled by $ε^D$, but for $K_{uu}^{-1}Σ_uK_{uu}^{-1}$ each element is scaled by $ε^{2D}$. Then the error in the means in $Mε^{2D}$. For the variance, the term in equation 61 is $ε^D ∑\_m (|c_m(x_*)|^2 - |c_m'(x_*)|^2$. But the summand is in $O(ε^2D)$ for every $m$ by Lemma C.1. Then the whole term is in $O(Mε^3D)$. For the terms in equations 62 and 63, we will recombine them. This gives $ε^{2D}∑_{mm'}Σ_{u,mm'} (c_m(x_*)c_{m'}^*(x_*)-c_m'(x_*)c_{m'}'^*(x_*))$ but by a minor adaptation of Lemma C.1, the summand is again in $O(ε^{2D})$, so the overall rate is $O(M^2ε^{4D})$. When $ε$ is set appropriately as a function of $M$ so that this is a decaying function of $M$, then for sufficiently large $M$ the term is in $O(Mε^{2D})$.
>
> (7) To clarify the first point, we rephrase. By variationally correct we mean inducing values $u_m$ which can a priori be viewed as a well-defined linear functional of the process $f$. In the second case we mean the stated result in equation 14 -- we will make the citation precise.
>
> (8) Agreed that it is not necessarily slower. But you're wasting learning effort in getting a good variational approximation as well as learning the parameters, but your objective is far cheaper to evaluate, and usually you end up using a first order optimiser (Adam). If you can evaluate the closed form objective sufficiently cheaply and combine it with a second order optimiser (L BFGS-B) you could go faster.
>
> (9) The parameters of the covariance function are also learnt hyperparameters (mentioned in background, after equation 3).
>
> (10) yes

---

### Comment · Action_Editors · 2023-10-09
**Update and Previous submission**

Dear authors, one of the reviewers dropped off for this paper so I assigned another one.  Thus the review process may be a little delayed.

In the meantime, the previous TMLR submission url you provided doesn't seem to work.  Was there a previous version of this paper submitted to the journal?

Thank you,

AC

---

> ### Comment · Action_Editors · 2023-12-19
> **Update on progress**
>
> Dear authors, I apologize for the ongoing delay in getting your reviews.  The replacement reviewer had some personal circumstances that have delayed their review unfortunately.  I have been assured this review should be done by the end of the month.  Meanwhile, I have added a fourth reviewer just in case, and to round out the reviews.  All the best,
>
> Your AE.

---

> ### Comment · Action_Editors · 2024-01-24
> **Author response**
>
> Apologies once again for the delays in getting the reviews.  All the reviews are in now and the discussion phase should have been triggered.
>
> - Your AE

---

> > ### Author Response · Authors · 2024-03-18
> > **Progress update?**
> >
> > Dear AE, the reviewers should have submitted their formal decision recommendation no later than 17 February, about one month ago. Will there be any further progress?

---

> > > ### Comment · Action_Editors · 2024-03-18
> > > **Yes**
> > >
> > > Dear authors, I just submitted the decision for approval.  Apologies again for the multiple delays on this one.  Unfortunately, this paper was quite unlucky in reviewers dropping out and delays with medical concerns.

---

### Author Response · Authors · 2024-02-01
**summary of revisions**

Thanks to all the reviewers for their helpful comments. We have uploaded the revised submission, incorporating changes requested by reviewers.

Major changes:
* We amended the assumption about the first derivative to be a relative bound, which will be satisfied by a broad range of covariance functions. We have made suitable modifications to Lemma 4.1 and the proofs of the theorems to suit the new assumption.
* We amend theorem 4.4 to explicitly state the result in terms of the spectral density's tail bound exponent $q$. An advantage of this formulation is that it does not depend explicitly on $D$. We also amend the result and proof to to deal with the issues and omitted details raised by reviewer bFVe.
* We have focused the first part of section 4 entirely on the one dimensional case for clarity. We separately address the higher dimensional generalisations in the new section 4.2. The main part of the results in theorems 4.2 and 4.4 are unchanged in the higher dimensional case.
* We explicitly enumerate the assumptions about the spectral density and the placement of the inducing frequencies. We give the 1D version in section 4.1, and the generalised version in 4.2. We add commentary on the (revised) first derivative assumption and the placement of inducing frequencies in response to comments from the reviewers. We reference the assumptions when used.

Minor:
* Corrected various typos and unclear phrasing.
* Made minor amendments to notation to make it more standard.
* Rephrased the introductory comments around stochastic optimisation with minibatching following comments from reviewers
* moved figure 1 earlier.

---

### Decision · Action_Editor_Jpvb · 2024-03-18

**Recommendation:** Accept as is

**Comment:**

This paper presents a method for speeding up Gaussian process modeling of spatial data, using what they call integrated variational features.  Previously there were methods that took advantage of the spatial structure of the data to reduce the computational complexity of GP inference from N^3 to M^3, where N and M are the number of examples and number of feature dimensions respectively.  However, those methods were restricted to a small restricted set of covariance functions, limiting the modeling flexibility.  This work extends the M^3 complexity to many more covariance functions, using the variational Fourier features.  This would allow for much more flexible modeling of large spatial datasets.

All three reviewers voted for accept (two leaning, one accept).  The reviewers found that the paper was well motivated, addressed an important and relevant problem, the methods were well justified and empirical results convincing.  One reviewer found that the theoretical results were somewhat confusing and unverifiable.  However, after reading over the author response, that reviewer now seems satisfied.  The reviewers all found that the claims made in the paper were well supported by the empirical and theoretical evidence.  Overall, this is a strong paper that advances the state-of-the-art in efficient spatial modeling using Gaussian processes and seems like a good candidate for acceptance to TMLR.  Therefore I recommend acceptance.

**Audience:**

Spatial modeling with Gaussian processes remains an important subject in machine learning.  Making modeling faster and more flexible, with more interesting covariance functions, is certainly of interest and could be quite useful both from a research perspective and for practitioners doing spatial modeling.

**Claims And Evidence:**

The reviewers all agreed that the claims in the submission were supported by theory and experiments.